# Maintenance of cell wall remodeling and vesicle production are connected in *Mycobacterium tuberculosis*

**Vivian C Salgueiro-Toledo[1†], Jorge Bertol[2†], Claude Gutierrez[3], Jose L Serrano-Mestre[1], Noelia Ferrer-Luzon[2], Lucia Vázquez-Iniesta[1], Ainhoa Palacios[4], Laia Pasquina-Lemonche[5], Akbar Espaillat[6], Laura Lerma[1], Brian Weinrick[7], Jose L Lavin[8], Felix Elortza[4], Mikel Azkargorta[4], Alicia Prieto[9], Pilar Buendía-Nacarino[10], Jose L Luque-García[10], Olivier Neyrolles[3], Felipe Cava[6], Jamie K Hobbs[5], Joaquín Sanz[2*], Rafael Prados-Rosales[1*]**

[1]Department of Preventive Medicine and Public Health and Microbiology, Universidad Autónoma de Madrid, Madrid, Spain; [2]Institute for Bio-computation and Physics of Complex Systems BIFI, Department of Theoretical Physics, University of Zaragoza, Zaragoza, Spain; [3]Institut de Pharmacologie et de Biologie Structurale, IPBS, Université de Toulouse, CNRS, UPS, Toulouse, Spain; [4]CIC bioGUNE, Basque Research and Technology Alliance (BRTA), Bizkaia Science and Technology Park, Derio, Spain; [5]Department of Physics and Astronomy, University of Sheffield, Sheffield, United Kingdom; [6]Department of Molecular Biology and Laboratory for Molecular Infection Medicine Sweden, Umeå Centre for Microbial Research, SciLifeLab, Umeå University, Umeå, Sweden; [7]Trudeau Institute, Saranac Lake, United States; [8]Bioinformatics Unit, Neiker-Tecnalia, Derio, Spain; [9]Department of Microbial & Plan Biotechnology, Centro de Investigaciones Biológicas Margarita Salas, Spanish National Research Council (CSIC), Madrid, Spain; [10]Department of Analytical Chemistry, Universidad Complutense de Madrid, Madrid, Spain

**\*For correspondence:**
jsanz@bifi.es (JS);
rafael.prados@uam.es (RP-R)

†These authors contributed equally to this work

**Competing interest:** The authors declare that no competing interests exist.

## eLife Assessment

In this **important** study, the authors investigate the biogenesis of extracellular vesicles in mycobacteria and provide several observations to link VirR with vesiculogenesis, peptidoglycan metabolism, lipid metabolism, and cell wall permeability. The authors have done a commendable job of comprehensively examining the phenotypes associated with the VirR mutant using various techniques. The evidence presented in the revised manuscript is **convincing** and creates several avenues for further research.

**Abstract** Pathogenic and nonpathogenic mycobacteria secrete extracellular vesicles (EVs) under various conditions. EVs produced by *Mycobacterium tuberculosis* (*Mtb*) have raised significant interest for their potential in cell communication, nutrient acquisition, and immune evasion. However, the relevance of vesicle secretion during tuberculosis infection remains unknown due to the limited understanding of mycobacterial vesicle biogenesis. We have previously shown that a transposon mutant in the LCP-related gene *virR* (*virR*^mut) manifested a strong attenuated phenotype during experimental macrophage and murine infections, concomitant to enhanced vesicle release. In this study, we aimed to understand the role of VirR in the vesicle production process in *Mtb*. We employ genetic, transcriptional, proteomics, ultrastructural, and biochemical methods to investigate

the underlying processes explaining the enhanced vesiculogenesis phenomenon observed in the *virR*^mut. Our results establish that VirR is critical to sustain proper cell permeability via regulation of cell envelope remodeling possibly through the interaction with similar cell envelope proteins, which control the link between peptidoglycan and arabinogalactan. These findings advance our understanding of mycobacterial extracellular vesicle biogenesis and suggest that these set of proteins could be attractive targets for therapeutic intervention.

## Introduction

*Mycobacterium tuberculosis* (*Mtb*) is one of the most successful human pathogens with the capacity to persist within the host establishing long-lasting infections. *Mtb* possesses multiple highly evolved mechanisms to manipulate host cellular machinery and evade the host immune system. Similar to other intracellular pathogens, *Mtb* releases proteins and lipids, that interfere with host cellular processes, via canonical or specialized secretion systems (*Majlessi et al., 2015*). How mycobacteria confined within phagosomes delivers virulence factors to other cellular compartments and/or into the extracellular milieu to affect the physiology of neighboring cells is currently a matter of investigation. Understanding the physiology of this phenomenon during the infection process might be critical to develop alternative therapies against tuberculosis (TB).

Like most forms of life, *Mtb* uses an alternative antigen release process based on extracellular vesicles (EVs) (*Prados-Rosales et al., 2011*). Mycobacterial EVs (MEVs) are membrane walled spheres of 60–300 nm in diameter, released by live bacteria in vitro and in vivo. MEVs contain iron scavenging molecules, immunologically active complex lipids and lipoproteins, and classical virulence factors such as the *Mycobacterium ulcerans* toxin mycolactone (*Lee et al., 2015*; *Marsollier et al., 2007*; *Prados-Rosales et al., 2014b*; *Prados-Rosales et al., 2011*). Several studies have clearly established a role of MEVs in immunomodulation and shown that MEVs deliver factors that impair macrophage effector functions, inhibit T cell activation, and modify the response of host cells to infection (*Athman et al., 2015*; *Athman et al., 2017*; *Prados-Rosales et al., 2011*; *Rath et al., 2013*). MEVs have also shown potential as biomarkers of *Mtb* infection (*Schirmer et al., 2022*; *Ziegenbalg et al., 2013*). Although the importance of MEVs has been recognized, hardly anything is known regarding both how such vesicles are exported across the mycobacterial cell wall and the molecular mechanisms underlying vesicle formation in mycobacteria. To date, several modulators of MEVs have been described. Our group identified two conditions leading to stimulation of vesicle production, namely: iron starvation (*Prados-Rosales et al., 2014b*) and the deletion of *virR* (Rv0431, <u>v</u>esiculogenesis and <u>i</u>mmune <u>r</u>esponse <u>R</u>egulator) (*Rath et al., 2013*). In addition, it was shown that the Pst/SenX3-RegX3 signal transduction system regulates MEVs production independent from VirR (*White et al., 2018*), suggesting an alternative mechanism of vesicle production. We have recently demonstrated that both iron starvation and deletion of VirR in *Mtb*, two hypervesiculating conditions, trigger the induction of the isoniazid-induced gene operon *iniBAC* (*Gupta et al., 2023*). It was determined that both IniA and IniC are dynamin-like proteins (DLP) presumably assisting MEV release through the fission of the cell membrane (*Gupta et al., 2023*; *Wang et al., 2019*). Understanding how defects in VirR lead to hypervesiculation through DLPs is important to provide mechanistic explanation for the vesiculation process in mycobacteria.

Several lines of evidence support the notion that VirR might be involved in maintaining cell wall integrity: (i) First, sequence analysis of the VirR protein indicates that it contains a conserved LytR_C domain (*Figure 1—figure supplement 1*), which is usually found in combination with another domain named LytR-Cps2A-Psr (LCP), and proteins carrying both domains have been associated with the remodeling of the cell wall in Gram-positive bacteria (*Hübscher et al., 2008*). *Mtb* has six genes encoding for LCP proteins, three with both LCP and LytR_C domains (Rv3267, Rv3484 and Rv0822), two with a solo LytR_C domain (VirR and Rv2700) and one protein with a solo LCP domain (Rv3840; *Figure 1—figure supplement 1*). The LCP family includes enzymes that transfer glycopolymers from membrane-linked precursors to peptidoglycan (PG) or cell envelope proteins and are central to cell envelope integrity (*Kawai et al., 2011*). In mycobacteria its was demonstrated that some members of the (LCP) family of proteins are responsible for the linkage between arabinogalactan (AG) and PG (*Grzegorzewicz et al., 2016*; *Harrison et al., 2016*). Rv3267, Rv3484 and Rv0822 appear to have overlapping functions in cell wall assembly, Rv3267 being the primary ligase (*Grzegorzewicz et al.,*

*2016*). Interestingly, while knocking down the gene encoding for the main AG-PG ligase (cg0847, *lcpA*) in *Corynebacterium glutamicum* leads to the release of outer membrane material to the extracellular medium (*Baumgart et al., 2016*), this phenotype is not observed in any knockout strain in the *Mtb* orthologs (*Grzegorzewicz et al., 2016*), probably reflecting the ability of the mutants to negatively regulate the biosynthesis of cell wall constituents in response to a decrease in ligase activity; (ii) Second, we and others have determined the increased susceptibility of the transposon *virR* mutant (*virR^mut^*) with respect to the WT (*Ballister et al., 2019*; *Rath et al., 2013*), and the mutant in the *virR* ortholog, Rv2700 (cell envelope integrity, *cei*) to cell wall targeting drugs such as meropenem (*Ballister et al., 2019*). Based on these results, we hypothesize that LytR_C solo domain proteins including VirR and Cei Rv2700 have critical roles in the maintenance of cell wall homeostasis and their absence provokes cell wall alterations leading to enhanced vesiculation.

In this work, we show that the absence of VirR leads to ultrastructural changes in the cell envelope as observed by cryo-electron microscopy (cryo-EM). These changes are compatible with enhanced permeability and enlargement of the PG layer, as measured by high-resolution atomic force microscopy (AFM). We demonstrate that VirR interacts with canonical LCP proteins and propose a model for such interaction. These results indicate that VirR is a central scaffold at the cell envelope remodeling process assisting the PG-AG linking process and that this function is important for vesicle production in *Mtb*.

## Results

### High EV production linked to the lack of VirR is associated with an altered cell envelope in *Mtb* in the absence of cell lysis

While Gram-negative bacteria can release outer membrane vesicles directly into the extracellular environment from their outermost compartment, Gram-positive bacteria and mycobacteria have thick cell envelopes surrounding the cytoplasmic membrane that acts as a permeability barrier. The mycobacterial cell envelope is composed of PG covalently linked to AG, which in turn is decorated with exceptionally long-chain mycolic acids (MA). These MA, together with intercalating glycolipids, form the unique mycobacterial outer membrane (OM) (*Kalscheuer et al., 2019*). Previous analysis of mycobacterial vesicle associated lipids showed predominantly polar lipids, consistent with the cytoplasmic membrane being the likely origin of the vesicles (*Prados-Rosales et al., 2011*). This is also true in EV produced by Fe-limited *Mtb* whose lipid content consists mainly of polar lipids and the lipidic siderophore, mycobactin (*Prados-Rosales et al., 2014b*). Given that these vesicles must traffic from their point of origin in the plasma membrane through the cell envelope, we hypothesized that local remodeling of the cell envelope is necessary for EV release, and that this process may be exacerbated in a *virR* mutant (*virR^mut^*). To test this, we examined the cell envelope structure of *virR^mut^* grown in minimal medium (MM), as well as wild type (WT) and *virR^mut^* complemented (*virR^mut^*-C) strains (*Rath et al., 2013*; *Figure 1*). Using cryo-electron microscopy (cryo-EM) on whole cells, which allows the study of mycobacterial cell surface-associated compartments in a close-to-native state (*Sani et al., 2010*), we could easily discern three layers comprising the mycobacterial cell envelope: (i) the outer membrane (OM); (ii) an intermediate layer composed of two sublayers (L1 and L2, previously assigned to the mycolate-PG-AG (mAGP) network *Sani et al., 2010*); and (iii) the cytoplasmic membrane (CM) (*Figure 1*). We observed a significant increase in the thickness of the overall cell envelope (OM-CM) in the *virR^mut^* strain compared to the WT. This difference was largely explained by the expansion of the CM-L1 layer, which appeared also more granulated, suggesting cell wall alterations closer to the cell membrane and indicating aberrant production of either AG, PG or both. Complementation of *virR* restored normal cell envelope structure (*Figure 1A–C*).

In Gram-negative bacteria increased vesiculation has been observed in conditions associated with either loss or maintenance of periplasm integrity (*McMillan and Kuehn, 2021*). Moreover, a novel mechanism of vesicle production common to both Gram-negative and Gram-positive bacteria has recently been described, in which a cell explosion phenomenon driven by bacteriophages leads to the formation of EVs in culture when bacteria are under genotoxic stress (*Nagakubo et al., 2021*). Consequently, we tested whether the enhanced vesiculation described in *virR^mut^* (*Rath et al., 2013*) could be linked to cell lysis events. To do this, we analyzed the presence of cytoplasmic IdeR, a transcription factor which is not present in EVs and is not detected in secreted proteins preparations, in culture

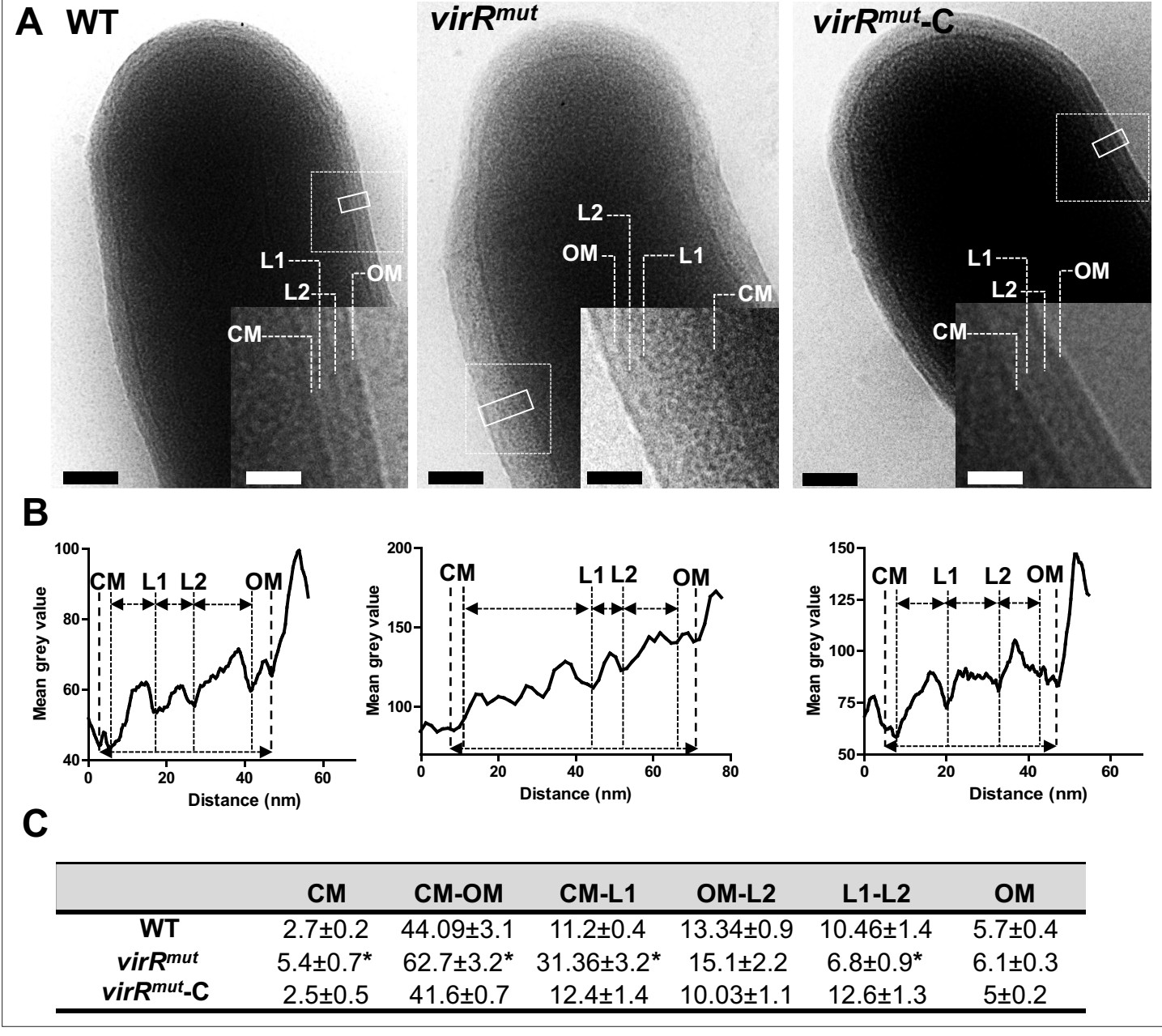

**Figure 1.** Ultrastructural changes in the cell envelope of *virR*-deficient mutant associated with increased vesiculation. (**A**) Cryo-electron micrographs of indicated *Mtb* strains grown in high iron MM. Closed line rectangles were used to calculate grey value profiles of cell envelopes using ImageJ. The dashed line insets within the main micrograph were magnified to show a detailed view of the cell surface. Scale bars are 100 nm in main micrographs and 50 nm in the insets. (**B**) Density profiles based on grey values of the cross sections marked by solid line rectangles in **A**. (**C**) Mean values and standard errors of distances in nm between main cell envelope layers measured in A and C. *$p<0.05$ after applying a Tukey´s multiple comparison test. The number of cells analyzed varied from n=20 (WT), n=15 (*virR^mut*) and n=23 (*virR^mut*-C). CM, cytoplasmic membrane; OM, outer membrane; L1, layer 1; L2, layer 2.

The online version of this article includes the following figure supplement(s) for figure 1:

**Figure supplement 1.** Diversity of *Mtb* LCP proteins.

supernatants of *Mtb* strains, assuming that its detection could be linked to a loss of envelope integrity (*Gupta et al., 2023*). We first confirmed that the *virR^mut* strain manifested an enhanced vesiculation by measuring EVs using nanoparticle tracking analysis (NTA; *Figure 2—figure supplement 1*). We did not detect differences in size distribution of isolated EVs between strains. The same supernatants

and cell lysates were used to detect the presence of IdeR by immunoblot using a specific polyclonal antibody (*Pandey and Rodriguez, 2014*). As a control for normal release of an extracellular protein, we used a monoclonal antibody against Ag85b (*Prados-Rosales et al., 2017*). We could not detect IdeR in the supernatant of WT and *virR^mut* strain, but a faint band corresponding to IdeR could be observed in the SN of the *virR* complemented strain (*virR^mut*-Comp). These results suggest that the absence of *virR* does not lead to cell lysis but its overexpression may compromise cell envelope integrity (*Figure 2—figure supplement 1*). We validated this approach by serially diluting WT, *virR^mut* and *virR^mut*-Comp strains on 7H10 solid media. As shown (*Figure 2—figure supplement 1*), we did not observe differences in the number of cells across the diluted spots between strains. Taken together, the observed changes in the cell envelope of high EV producing bacteria suggest that cell envelope alterations in response to lack of VirR and increased membrane vesicle biogenesis may be connected in *Mtb*.

## The absence of LytR_C solo domain proteins in *Mtb* leads to enhanced cell envelope permeability concomitant to hypervesiculation

The collective cryoEM observations that indicated an enlargement of the compartment next to the cell membrane may be linked to a more permeable cell envelope. Supporting this notion is the recent finding that *virR^mut* or the mutant in the *virR* ortholog *cei* (Rv2700) manifest an enhanced uptake of ethidium bromide (EtBr) relative to WT strain (*Ballister et al., 2019*). We could reproduce these results and add that *virR^mut*-Comp strain restores the permeability levels of the WT strain (*Figure 2A*). This enhanced permeability was concomitant to an increased sensitivity to different antitubercular drugs, including meropenem and vancomycin (*Ballister et al., 2019*). In agreement with these results, we could measure significantly enhanced sensitivity of *virR^mut* to the PG-targeting drug vancomycin relative to WT and *virR^mut*-Comp strains (*Figure 2B*). Interestingly, the magnitude of the sensitivity to cell wall targeting drugs was either moderate (ethambutol) or absent (isoniazid) in *virR^mut* (*Ballister et al., 2019*).

Furthermore, we observed enhanced incorporation of fluorescent D-amino acids (FDAAs) in *virR^mut* relative to the WT and complemented strains (*Figure 2C*). These probes can freely enter cells and may be incorporated into PG by at least three different mechanisms, depending on the species: through the cytoplasmic steps of PG biosynthesis and via two distinct transpeptidation reactions taking place in the periplasm (*Kuru et al., 2019*). Consequently, the differential labeling observed in *virR^mut* relative to WT strain may be a consequence of altered PG turnover in the mutant.

We then tested whether the enhanced permeability and vesiculation of *virR^mut* are also connected in an *Mtb* mutant lacking *cei* (*Ballister et al., 2019*). To do this, we generated *cei* conditional mutants using the CRISPR interference (CRISPRi) technology (*Rock et al., 2017*). Further, we also generated *virR* CRISPRi conditional mutants to properly compare both strains (*Figure 2D*). When these strains were cultured in the presence of anhydrotetracycline (ATc) to induce the transcriptional silencing of target genes, we could measure enhanced vesiculation by analyzing vesicle concentration in the supernatant by NTA (*Figure 2E*), relative to strains cultured in the absence of ATc. Overall, these results validate the enhanced permeability phenotype of *Mtb* mutants in the LytR_C solo domain proteins VirR and Cei and establish that this subfamily of proteins participates in the regulation of vesicle production in *Mtb*. Moreover, the higher susceptibility to PG-cross-linking inhibitors in both mutants and the enhanced incorporation of FDAAs in the *virR* mutant strain suggests defects in PG remodeling.

## Transcriptional profiling of *virR^mut* showcases systemic alterations in cell wall architecture and metabolism

To identify *virR*-dependent mediators of vesicle production, we carried out a transcriptional profiling of *virR^mut* and compared it to those from a WT and *virR^mut*-Comp strains. Previous efforts to define the transcriptional profile of *virR^mut* showed no significant differences compared to WT (*Rath et al., 2013*). However, those experiments were carried out with bacteria grown in rich 7H9 medium, while here we use a defined high-Fe minimal medium (MM), which is typically used for vesiculation experiments (*Prados-Rosales et al., 2011*). We extracted RNA from four independent cultures (average optical densities (OD) of 0.40, 0.38, and 0.42 for WT, *virR^mut*, and *virR^mut* -Comp, respectively) 1 day before vesicle isolation, and conducted RNA-seq (*Figure 3A*). Principal component analysis (PCA)

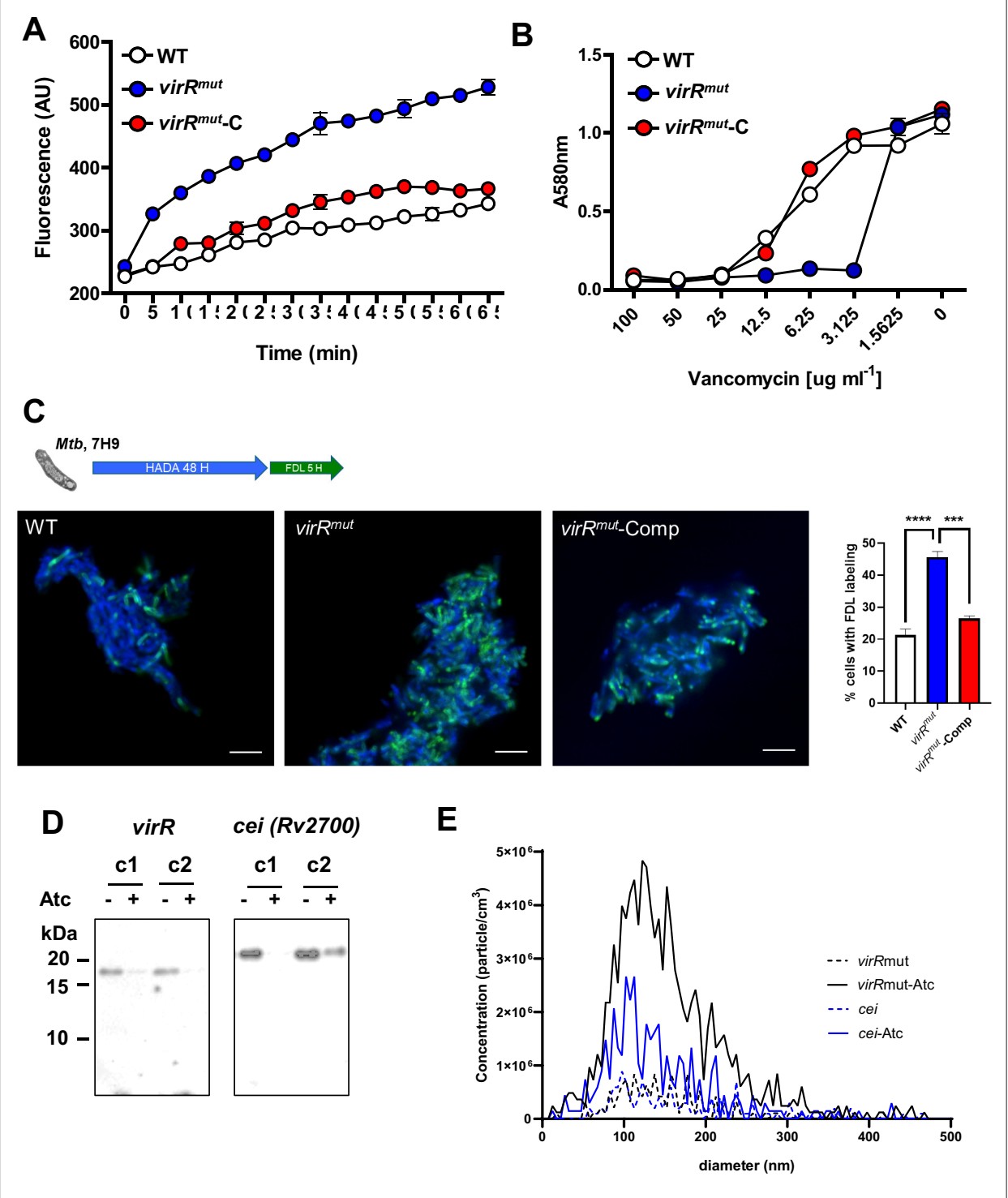

**Figure 2.** Enhanced permeability and vesiculation are linked in the absence of LytR_C solo domain proteins in *Mtb*. (**A**) Uptake of ethidium bromide (EthBr) in the indicated strains, as measured by fluorescence at 590 nm for 65 min. Data are mean and standard deviation of three biological replicates. (**B**) Sensitivity of the indicated strains to different concentrations of vancomycin as measured by monitoring optical density at 580 nm (OD580) after 7 days. Data are mean and standard deviations of three biological replicates. Errors bars not shown are smaller than symbols. (**C**) Time course of the incorporation of FDAAs on the indicated strains in 7H9 as measured by fluorescence. Cells were labeled with HADA for 48 hr, washed in 7H9 medium and labeled with FDL for 5 hr. Then, cells were fixed and imaged under a fluorescence microscope. Representative images of WT, *virR*^*mut*^ and *virR*^*mut*^-Comp strains in 7H9. Data are mean and standard deviations of three biological replicates. *** means p<0.001 and **** means p<0.0001 after applying a One-way ANOVA analysis. (**D**) immunoblot analysis of cell lysates of two independent conditional mutants (**c1 and c2**) for *virR* and *cei* for the presence

*Figure 2 continued on next page*

*Figure 2 continued*

of VirR and Cei proteins in cultures incubated with and without anhydrotetracycline (ATc). (**E**) Nanoparticle tracking analysis (NTA) of EV preparations derived from *virR* and *cei* conditional mutants (c2 (*virR*) and c1 (*cei*)) obtained from cultures with and without ATc showing number of particles per cm³ determined by Zeta View NTA in three independent EV preparations. Data are presented as mean ± SEM.

The online version of this article includes the following source data and figure supplement(s) for figure 2:

**Source data 1.** Png containing original immunoblot images, indicating the relevant blots and strains, displayed in *Figure 2D*.

**Source data 2.** Original files for immunoblot analysis displayed in *Figure 2D*.

**Figure supplement 1.** The enhanced vesiculation phenotype of *virR^mut* occurs in the absence of cell lysis.

**Figure supplement 1—source data 1.** Png containing original immunoblot images, indicating the relevant blots and strains, displayed in *Figure 2— figure supplement 1B*.

**Figure supplement 1—source data 2.** Original files for immunoblot analysis displayed in *Figure 2—figure supplement 1B*.

**Figure supplement 2.** Quantitative RT PCR of *virR* downstream genes Rv0432 and Rv0433.

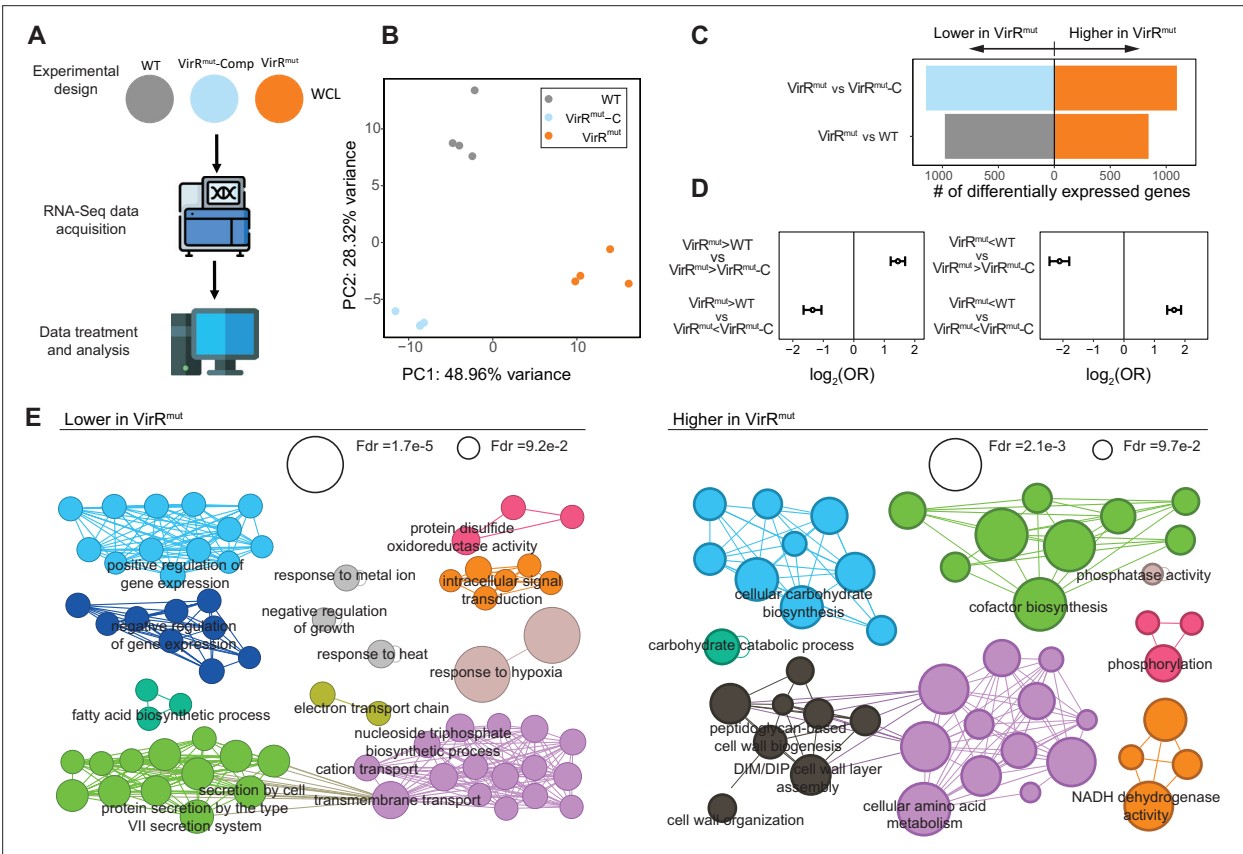

**Figure 3.** Transcriptional profiling of *virR^mut* showcases systemic alterations in cell wall architecture and metabolism. (**A**) Experiment design for RNA-seq analyses. RNA was extracted from whole cell extracts from four replicate cultures of each strain (WT, *virR^mut*, and *virR^mut* -Comp), and sequenced. (**B**) PCA plot of transcriptomics data. The loadings from the two first principal components are plotted. (**C**) Number of differentially expressed genes between *virR^mut* and either WT (left) or *virR^mut*-Comp (right). (**D**) Enrichment profiles between sets of genes found up and down-regulated for the contrasts *virR^mut* vs WT and *virR^mut* vs *virR^mut*-Comp. While genes that are differentially expressed in the same direction in both contrasts are mutually enriched, genes whose differential expression goes in different directions in both comparisons are mutually depleted (two-sided Fisher exact test p<0.05 in all four cases). (**E**) Gene Ontology (GO) enrichments of genes that are (1) simultaneously downregulated in *virR^mut* with respect to both WT and v *virR^mut*-Comp (left) and (2) simultaneously upregulated (right). GO terms selected for testing included biological process and cell compartment labels containing N>10 genes, between levels 4–6 of the ontology tree. In this representation, each node in the networks represents a significantly enriched term (terms selected for visualization with FDR = 10% and enrichment odds-ratio >2), nodes' size is proportional to the enrichment significance, and links between terms are added wherever the sets of genes contributing to the enrichments in a connected pair show an intersection larger than 50% of the smallest of the gene sets involved in the pair (see Materials and methods for further details, and *Supplementary file 2* for an extended list of all the enrichments found).

was then performed on the transcriptomic data, after filtering out lowly expressed genes, as well as genes containing outlier observations (N=3985 genes tested, see Materials and methods), and revealed significant differences between strains. The first principal component, explaining 49% of the variance, separated *virR^mut^* strain samples from strains bearing functional copies of *virR* (*Figure 3B*, WT and *virR^mut^*-Comp vs *virR^mut^*, one-tailed Mann-Whitney test p=2.02E-2). Furthermore, the second principal component (28.3% variance) captured *virR*-independent effects of mutagenesis on transcriptional profiles, as it separated WT samples from *virR^mut^*-Comp and *virR^mut^* (one-tailed Mann-Whitney test p=2.02E-2).

After PCA, we conducted differential expression analysis between strains to find 1808, and 2230 genes differentially expressed in *virR^mut^* compared to either WT, or *virR^mut^*-Comp strains, respectively (*Supplementary file 1*, Benjamini-Hochberg FDR <0.05), representing 45.4% and 56.0% of all genes tested. Although transcriptional profiles of the WT and *virR^mut^*-Comp strains are also significantly diverse, with up to 2184 differentially expressed genes between them, the differences in expression between *virR^mut^* when compared to either WT or *virR^mut^* -Comp are significantly correlated (r=0.63, p<2.2e-16), and the sets of both upregulated and downregulated genes in *virR^mut^* vs any of the other strains are mutually, directionally enriched (*Figure 3D*). Finally, we defined a set of N=399 genes that are simultaneously upregulated in *virR^mut^* with respect to any of the two strains containing functional copies of *virR* (logFC >0 and FDR <0.05 for both *virR^mut^* vs WT and *virR^mut^* vs *virR^mut^*-Comp), as well as a set of N=502 genes downregulated in *virR^mut^* vs both *virR*-containing strains. This way, the differential expression signature found for these genes in the *virR^mut^* strain is most likely VirR-dependent and not-induced by non-specific consequences of the mutagenesis procedures.

Next, we looked for gene ontologies (GO) enriched among these *virR*-dependent genes, stratified by the direction of the effects (399 up-regulated, and 502 down-regulated in *virR^mut^* relative to WT and *virR^mut^*-Comp alike). A total of 51 GO terms related with biological processes and cell compartment appeared significantly enriched among genes upregulated in the *virR^mut^* strain (FDR = 10%), while 85 terms were enriched among downregulated genes (FDR = 10%, see Materials and methods, and *Supplementary file 2*). When focusing on the most prominent of these enrichments (OR >2), the analyses highlighted several groups of biological processes and cell compartment ontologies that appeared altered in *virR^mut^* with respect to both WT and *virR^mut^*-Comp strains (*Figure 3E*).

First, among genes that were expressed at lower levels in *virR^mut^*, we found terms related to several stresses associated with the intra-phagosomal environment (*Schnappinger et al., 2003*), such as response to hypoxia (FDR = 1.7E-5), heat (FDR = 2.2E-2) or response to metal ions (FDR = 3.4E-2), suggesting that responses to host-induced stress are compromised in the mutant, as it is also suggested by the weaker, albeit significant enrichments of responses to host immune response (OR = 1.88, FDR = 7.47e-02, not highlighted in *Figure 3E*). Furthermore, the mutant strain showed a reduction in transcripts related with secretion (FDR = 7.6e-03), and, specifically, protein secretion by the type VII secretion system (FDR = 5.2 e-03). Contributing to this enrichment, we found that the key antigen ESAT-6 was significantly down-regulated in *virR^mut^* with respect to both WT (log2FC = −0.17, FDR = 4.1E-3) and *virR^mut^*-Comp strains (log2FC = −0.19, FDR = 2.6E-3), while *cfp10* (Rv3874), was similarly less expressed in *virR^mut^* than in WT (log2FC = −0.24, FDR = 2.6E-3).

Second, we found enriched terms related with lipid metabolism and transport both among the sets of genes up and downregulated in the *virR^mut^* strain. On the one hand, genes downregulated in *virR^mut^* appeared enriched in fatty acid biosynthetic processes (FDR = 6.7E-2), suggesting that the regulation of both anabolic and catabolic pathways of lipids, key for bacterial survival and pathogenesis upon infection (*Ghazaei, 2018*), are, at least in part, VirR-dependent. Furthermore, the mutant strain also shows a significant induction of a number of GO terms related with cell wall composition (e.g. cell wall organization: FDR = 3.9e-2), including terms associated with the biosynthesis key components of the mycobacterial cell wall such as PDIMs (DIM/DIP cell wall layer assembly enrichment FDR = 9.5e-03) and importantly peptidoglycan (peptidoglycan-based cell wall biogenesis FDR = 2.2e-2) (*Figure 3F*).

Third, we found a series of enrichments related with global regulation of gene expression (positive: FDR = 4.4e-2, negative: FDR = 5.4e-2), and RNA biosynthesis (positive: FDR = 5.4e-2, negative: FDR = 5.8e-2), among genes downregulated in *virR^mut^*, suggesting a global alteration of the transcriptional landscape in the mutant.

Fourth, we found that metabolic reprogramming in the *virR^mut^* strain is not restricted to lipids but also involves amino acids (cellular amino acid metabolic processes FDR = 2.1e-3), and carbohydrates

(cellular carbohydrate biosynthetic process FDR = 6.8e-3), enriched among genes upregulated in the mutant. Similarly, energy production also appears altered in *virR^mut*, with related GO terms enriched among downregulated genes, such as electron transfer activity, (FDR = 4.2e-2), as well as terms enriched among upregulated genes in the mutant such as NADH dehydrogenase activity, (FDR = 3.3e-3). All together, these results combined with the differences concerning lipid metabolism discussed above, suggest a global remodeling of metabolic networks in *virR^mut*.

## *virR*-dependent genes intersect the regulons of key transcriptional regulators of responses to stress, dormancy, and cell wall remodeling

To shed light on the specific transcriptional circuits whose disruption in the *virR^mut* strain may lead to the differential gene expression patterns described, we resorted to the regulon annotations previously reported (*Minch et al., 2015*). In this work, authors collected ChIP-seq data on binding events between a panel of transcription factors (TFs) and DNA to annotate interactions between 143 regulators and the promoters of 7248 target genes (peaks between −150 and +70 nucleotides from transcription staring sites). Capitalizing on these data, we interrogated whether the sets of genes up- (N=399) and downregulated in the *virR^mut* strain (N=502) were significantly enriched among each of the regulons reported by Minch et al. A total number of 10 regulons appeared enriched among either down (9) or upregulated genes (1) in *virR^mut* (*Figure 4A*, *Supplementary file 3*, highlighted TFs at OR >2 FDR <0.1). Furthermore, the expression of 7 out of 10 of the TFs controlling these regulons was itself found to be *virR*-dependent in our transcriptional data (*Figure 4B*). Genes more highly expressed in the *virR^mut* strain appeared significantly enriched among Rv0339 targets (OR = 9.1, FDR = 7.5E-2, *Figure 4A*), named IniR because of its role as key regulator of the isoniazid-induced operon *iniBAC*, a key element of the response to cell wall biosynthesis inhibition induced by isoniazid (*Boot et al., 2017*) and linked to vesicle biogenesis in *Mtb* (*Gupta et al., 2023*). Several additional transcription factors known to regulate cell wall biosynthesis and homeostasis were also enriched among genes downregulated in the *virR^mut* strain. These include SigF (OR = 5.8, FDR = 8.1E-2), a sigma factor controlling cell envelope organization (*Williams et al., 2007*), *mtrA* (OR = 2.2, FDR = 6.0E-2), a TF controlling peptidoglycan cleavage and alterations in the context of regulation of cell growth and division, whose disruption has recently been shown to lead to increased permeability (*Peterson et al., 2023*); as well as Rv0472c (OR = 11.7, FDR = 3.5E-2), a transcription factor involved in the regulation of mycolate remodeling pathways (*Peterson et al., 2019*), a process used by the pathogen upon phagocytosis to evade macrophage-induced stress and enter dormancy. Interestingly, two genes under the control of Rv0472c, that are key to enable this process, are the mycolate desaturases *desA1* and *desA2* (*Peterson et al., 2019*), which appear downregulated in the *virR^mut* strain with respect to WT (logFC = −0.4,–0.1; FDR = 2.3E-11, 3.2E-2, for *desA1*, and *desA2*, respectively), as well as with respect to *virR^mut*-Comp (logFC = −0.4,–0.4; FDR = 7.2E-12, 5.3E-21). Furthermore, Rv0472c has also been linked with the regulation of key proteins in the response to several sources of stress in *Mtb*, such as the putative nitroreductase Rv3131 (response to oxidative/nitrosative stress), as well as the triacylglycerol synthase *tgs1*, responsible for the synthesis and accumulation of triacylglycerol, a major energy source for the bacterium during dormancy (*Chauhan and Tyagi, 2009*), both of which are strongly downregulated in the *virR^mut* strain vs both WT: (logFC = −1.2,–1.7; FDR = 3.1E-19, 3.2E-14) and *virR^mut*-Comp (logFC = −0.4,–1.9; FDR = 3.9E-5, 1.4E-17).

The adjacent genes Rv3131 and *tgs1* (Rv3030c) appear under the control of other key regulators of the response to intracellular stress in *Mtb* and the transition to dormancy, whose regulons appear as well enriched among genes downregulated in *virR^mut* (*Figure 4C*). These include *devR* (OR = 2.7, FDR = 1.9E-5), the canonical regulator of the response to hypoxia and transition to dormancy in *Mtb* (*Park et al., 2003*), as well as Rv1985c (OR = 4.1, FDR = 4.5E-11), a key regulator of the homeostasis of the threonine biosynthesis pathway, a metabolic route needed by the bacterium for persistence in vivo (*Hasenoehrl et al., 2019*). Three of the four remaining TFs whose regulons we found enriched among *virR^mut* downregulated genes have been similarly linked to the response to assorted sources of stress that are cardinal features of the phagosome environment, such as oxidative stress, response to toxic metals, and low pH (*Schnappinger et al., 2003*). These include the regulator *lsr2* (OR = 2.3, FDR = 5.1E-4), a global histone-like protein responsible from protecting the DNA physically from oxidative stress (*Bartek et al., 2014*), the zinc-uptake regulator Zur (*furB*) (*Maciag et al., 2007*) (OR = 8.8, FDR = 4.5E-2), and *tcrX* (OR = 2.6, FDR = 5.0E-2), part of the recently characterized acid-sensing

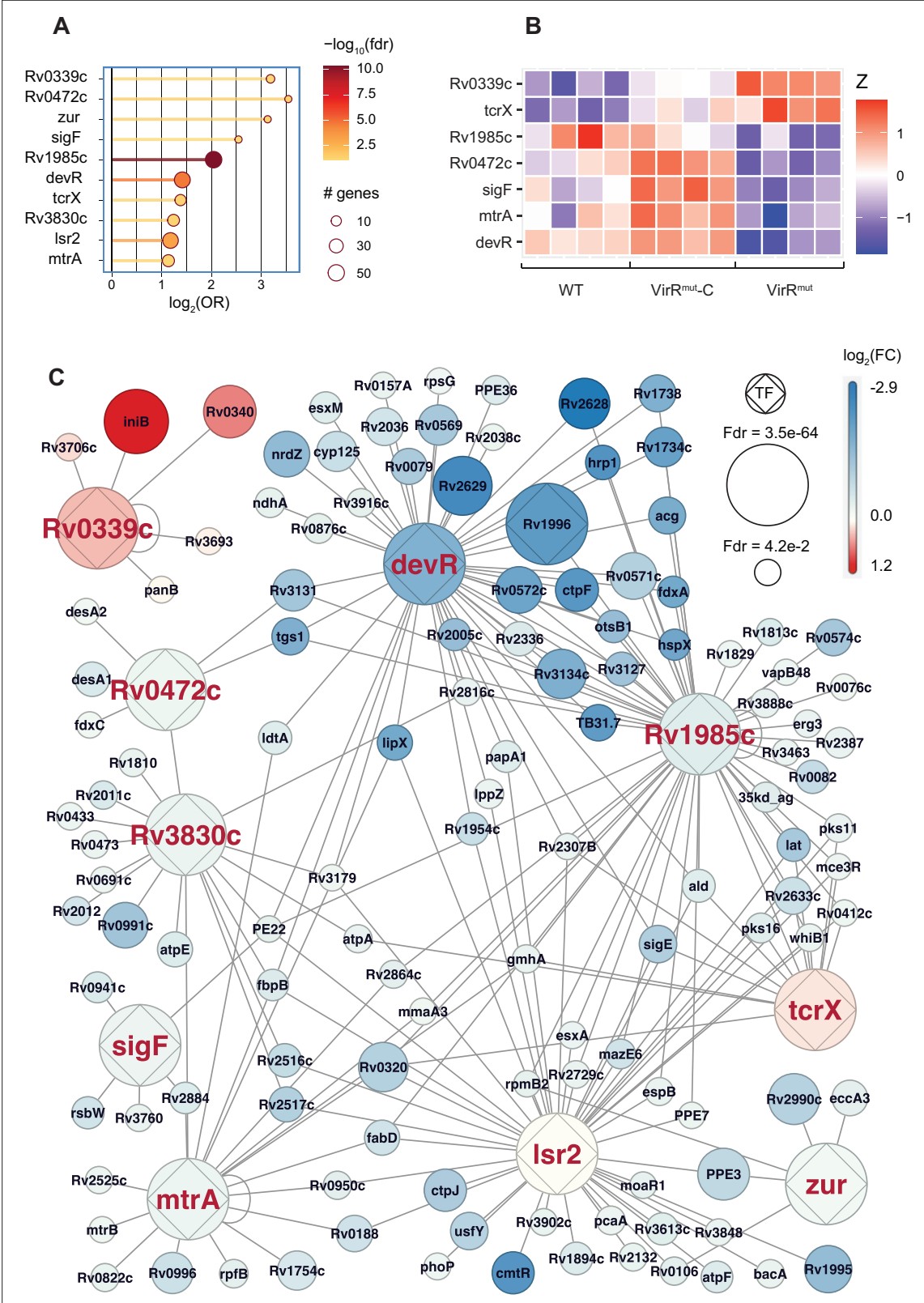

**Figure 4.** VirR-dependent genes intersect the regulons of key transcriptional regulators of the responses to stress, dormancy, and cell wall remodeling. (**A**) Transcription factors controlling regulons significantly enriched among genes that are (1) simultaneously downregulated in *virR*^mut^ with respect to both WT and *virR*^mut^-Comp (bottom, 9 TFs) and (2) simultaneously upregulated (top, 1 TF). (**B**) Differential expression patterns for enriched TFs whose

*Figure 4 continued on next page*

*Figure 4 continued*

expression is VirR-dependent. (**C**) Regulatory subnetwork including the genes under transcriptional control of the enriched TFs highlighted in **A** that were found to be VirR-dependent. In this visualization, color captures the logFC for the contrast *virR^mut* -WT, and the size of its FDR.

The online version of this article includes the following source data and figure supplement(s) for figure 4:

**Figure supplement 1.** $H_2O_2$ sensitivity assay.

**Figure supplement 1—source data 1.** Png containing original images for the $H_2O_2$ sensitivity assay, indicating strains and dilution, as displayed in *Figure 4—figure supplement 1*.

**Figure supplement 1—source data 2.** Original files for the $H_2O_2$ sensitivity assay displayed in *Figure 4—figure supplement 1*.

two-component system tcrXY (*Stupar et al., 2024*). Notably, we observed that removing catalase from culture medium prior to submitting *virR^mut* to a source of oxidative stress like H2O2 for 8 hr, significantly affected its growth relative to WT and *virR^mut*-C strains (*Figure 4—figure supplement 1*). This result validates the transcriptomics data reflecting a defect in regulating oxidative stress in *virR^mut* and encourage to investigate its connection with vesicle production.

## Cell envelope permeability is a major driver of MEV release

If enhanced cell permeability is a global consequence of *virR* disruption that leads to enhanced vesiculation, we hypothesized that other conditions either inducing or decreasing cell envelope permeability could lead to alterations in vesicle production in *Mtb*. To test this, we treated *Mtb* with sublethal concentrations of thioridazine (TRZ), a known enhancer of cell permeability (*de Keijzer et al., 2016*); and, alternatively, we cultured the bacillus with cholesterol as a sole carbon source, a condition which reduces mycobacterial cell permeability (*Brzostek et al., 2009*; *Figure 5A*). The enrichment of the DevR regulon among the genes that are downregulated in *virR^mut* is also suggestive of the existence of a convergent regulatory route controlling both cell permeability and vesiculation under control of this transcription factor, whose activation has been described as a key part of the pathogen's transcriptomic response to cholesterol (*Soto-Ramirez et al., 2017*), concomitant to a reduction in cellular permeability (*Brzostek et al., 2009*). Importantly, neither cholesterol nor TRZ altered bacterial viability as measured by the absence of the release of the transcription factor IdeR, as an indication of cell lysis or by spotting serially diluted cultures on solid medium from the corresponding conditions (*Figure 5—figure supplement 1*). We then measured vesicle levels in culture supernatants and found that *Mtb* significantly releases more EVs upon treatment with TRZ, and significantly reduces vesicle release in the presence of cholesterol as a sole carbon source (*Figure 5B*), as measured by dot blot after normalization to colony forming units (CFUs). These results strongly suggest that cell envelope permeability determines the magnitude of the vesiculation phenomenon in *Mtb*. We next investigated whether common transcriptional features underpin the regulation of cell envelope permeability in the tested conditions. To do that, we compared the transcriptional response of *virR^mut* with that of available datasets including *Mtb* cultured in the presence of 0.01% of cholesterol as a sole carbon source (*Pawełczyk et al., 2021*), and *Mtb* treated with 6 µg ml-1 of TRZ (*Dutta et al., 2010*). Experimentally, we retrieved the set of genes up and downregulated in *Mtb* grown in cholesterol vs glucose-rich culture media (*Brzostek et al., 2009*), and interrogated whether those genes were enriched among the sets of differentially expressed genes in *virR^mut* vs WT, and *virR^mut* vs *virR^mut*-Comp reported in this study (*Figure 5C*). Interestingly, we found that those genes whose expression changed in the same direction in conditions associated with low, or high permeability, in both studies were mutually enriched, while the cross-comparisons between sets with opposite responses to low-permeability-associated treatments in both studies led to significant depletion statistics. These results validate the initial hypothesis according to which changes in permeability induced by these two treatments are linked to the activation of a common transcriptional program. The inspection of genes that are simultaneously upregulated in treatments associated with low permeability in both studies identified DevR as a putative regulator associated with the orchestration of this transcriptional footprint of low cell permeability. The two-component system DevS-DevR (also known as DosS-DosR) has a well-known role as a global regulator of the transition to dormancy, and its regulon is consistently upregulated in non-permeable dormant bacteria (*Honaker et al., 2009*).

Interestingly, authors found that low-permeability bacteria cultured in a cholesterol medium, albeit featuring a regular growing phenotype, activated the DosR regulon, suggesting that cell wall

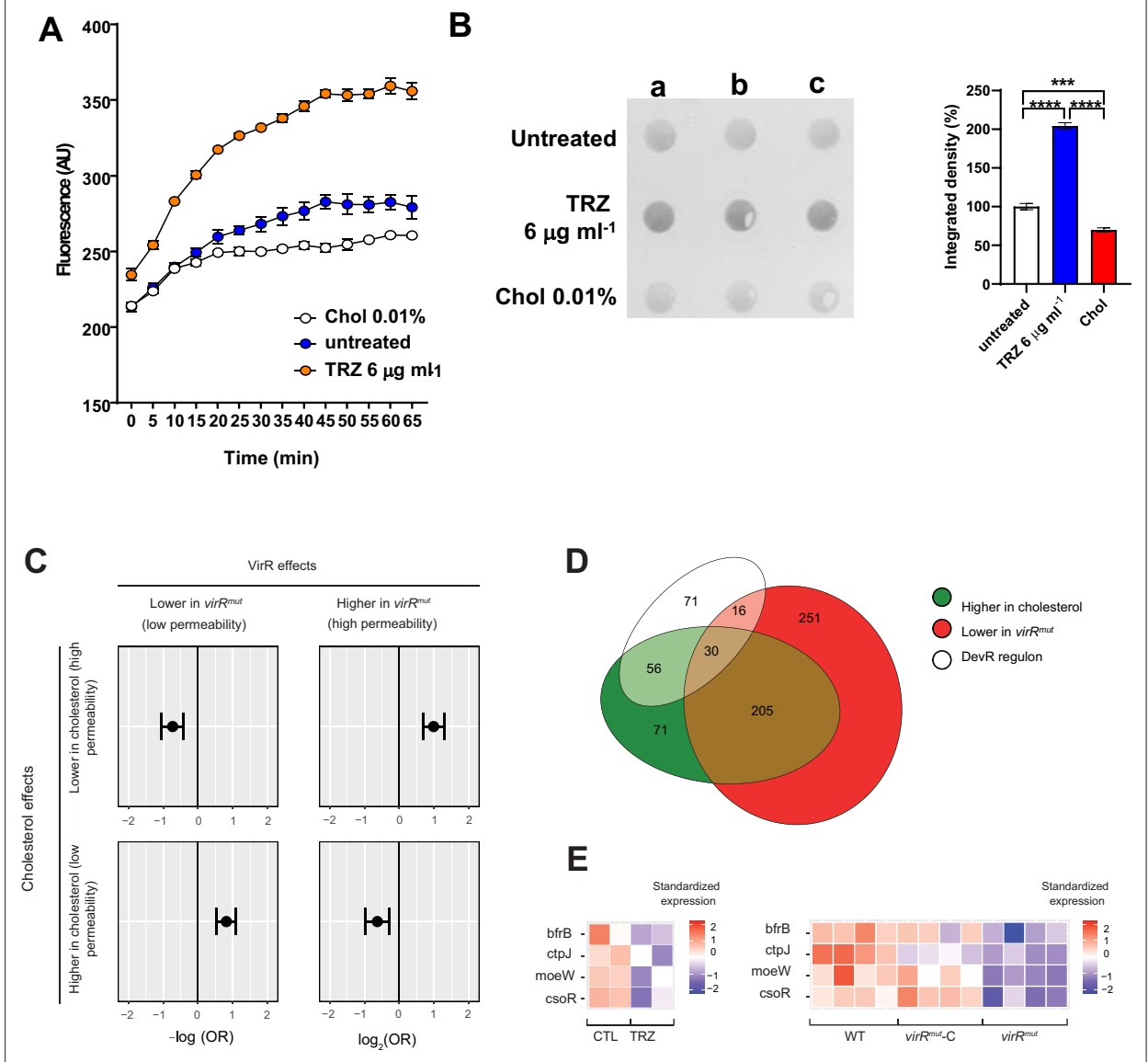

**Figure 5.** Cell permeability is a major driver of vesicle production in *Mtb*. (**A**) Time course of the uptake of ethidium bromide (EthBr) by *Mtb* grown in the indicated conditions, as measured by fluorescence at 590=nm for 65 min. Data are mean and standard deviation of three biological replicates. (**B**) Dot-blot analysis of released EVs in supernatants from *Mtb* cultures submitted to the indicated conditions. The loading volume was normalized according to CFUs. The graph below represents the quantification of the dotblot using ImageJ. The integrated cell density was normalized to WT values. Data are mean and standard errors from the three biological replicates; **** denotes p<0.0001, *** denotes p=0.0028 after applying a Tukey's multiple comparison test. (**C**) Enrichment odds ratios for Fisher exact tests comparing sets of genes up, and down regulated, respectively, in (1) cholesterol-supplemented medium vs. glucose-supplemented medium (*Pawełczyk et al., 2021*), GEO accession GSE175812; and (2) *virR^mut* vs WT as well as *virR^mut* vs *virR^mut*-Comp. (**D**) Euler diagram showing the intersection between (**A**) genes upregulated by cholesterol, (**B**) genes down-regulated in *virR^mut*, and (**C**) the DevR regulon. (**E**) Expression patterns in response to TRZ (1 hr) and in *virR^mut* for genes involved in metal signaling, metabolism, and homeostasis.

The online version of this article includes the following source data and figure supplement(s) for figure 5:

**Source data 1.** Png containing original Dot-blot analysis images, indicating the relevant conditions, displayed in *Figure 5B*.

**Source data 2.** Original files for immunoblot Dot-blot analysis displayed in *Figure 5B*.

**Figure supplement 1.** Control of cell lysis and cell viability change under conditions that affect permeability.

**Figure supplement 1—source data 1.** Png containing original immunoblot images, indicating the relevant blots displayed in *Figure 5—figure supplement 1A*.

**Figure supplement 1—source data 2.** Original files for the immunoblot assay displayed in *Figure 5—figure supplement 1A*.

*Figure 5 continued on next page*

remodeling leading to a decrease in permeability linked with the induction of DosR can be triggered independently to the transcriptional regulatory programs leading to growth arrest and dormancy in *Mtb* (*Pawełczyk et al., 2021*). Importantly, the integration of genes that are downregulated in *virR^mut^* validates this hypothesis (*Figure 5D* and *Figure 5—figure supplement 2*), since not only the intersection between the DevR regulon and the set of genes downregulated in *virR^mut^* are significantly larger than expected by chance (see *Figure 4A*, OR = 2.67, p=2.7E-7), but also the intersection between DevR regulon and the genes that are activated by cholesterol in *Brzostek et al., 2009*. data (OR = 1.89, p=3.5e-5). Thirty out of the 173 genes annotated in the DosR regulon described in *Minch et al., 2015* are simultaneously upregulated in conditions linked to low permeability in both studies.

A similar analytic approach for the comparison of the response of *virR^mut^* and the response to the drug TRZ, a treatment well-known for inducing an increase in permeability in *Mtb*-treated cells (*Dutta et al., 2010*; GSE16626), led to conceptually similar results, with genes simultaneously downregulated in both TRZ and *virR^mut^* (OR = 1.96, p = 9.1E-4) being enriched. Interestingly, among the 22 genes showing the largest, and most significant repression effects by both treatments (log2FC >0.25 and FDR <0.05 simultaneously, for TRZ effects, *virR^mut^* vs WT), we found four genes whose functions are related with metal nutrients uptake and homeostasis, such as genes encoding the bacterioferritin BrfB, the metal exporter CtpJ, the cupper regulator CsoR and Moew, involved in molibdenum-cofactors biosynthesis (*Figure 5E*). Taken together, these analyses highlight the transcriptional parallels between *virR^mut^* and both TRZ and cholesterol conditions, which strongly suggests the existence of a common transcriptional signature associated with the control of cell permeability.

## Modulation of EV levels and permeability in *virR^mut^* by cholesterol and TRZ

We next wondered about the effect of culturing *virR^mut^* on both cholesterol or TRZ could have on cell growth, permeability and EV production. In the case of cholesterol, it has also been shown to affect other aspects of physiology (redox, respiration, ATP), which can directly affect permeability (*Lu et al., 2017*). We monitored *virR^mut^* growth in MM supplemented with either glycerol, cholesterol as a sole carbon source, or TRZ at 3 μg ml$^{-1}$ for 20 days. While cholesterol significantly enhanced the growth *virR^mut^* after 5 days relative to glycerol medium, supplementation of glycerol medium with TRZ restricted growth during the whole time-course (*Figure 5—figure supplement 3A*). The study of cell permeability in the same conditions indicated that the enhanced cell permeability observed in glycerol MM was reduced when *virR^mut^* when cultured with cholesterol as sole carbon source. Conversely, the presence of TRZ increased cell permeability relative to the medium containing solely glycerol (*Figure 5—figure supplement 3C*). As we have previously observed for the WT strain, either condition (Chol or TRZ) also modified vesiculation levels in the mutant accordingly (*Figure 5—figure supplement 3B*). Interestingly, TLC analysis of cell membrane polar lipids showed that *virR^mut^* grown in cholesterol restores the previously observed higher Ac$_2$PIM$_2$ levels relative to WT when growing in glycerol (*Figure 5—figure supplement 4*). These results strongly indicate that other aspects of mycobacterial physiology besides permeability are also affected in the absence of VirR and may contribute to the observed enhanced vesiculation.

## *virR* regulates the protein content of EVs in *Mtb*

To capture the impact of transcriptional adaptation to the proteome, and the degree of divergence between transcriptional shifts and proteins changes observed in the absence of *virR*, we next performed label-free peptide mass spectrometry (LF-MS) of whole cell lysates (WCLs) of *virR^mut^*, WT, and *virR^mut^*-Comp strains. Additionally, we isolated EVs from axenic cultures and obtained their proteomic composition to examine for potential VirR-dependent differences in protein cargo that may alter the functionality of EVs (*Figure 6A*). As a result, we obtained proteomic profiles across strains and cell compartments for a total of N=935 proteins. Upon PCA analysis of the proteomic data (*Figure 6B*), we found that the first principal component (PC1), explaining 36.9% of total variance, separated samples from WCLs and EVs in all strains (PC1 differs across cell compartments: one-tailed Mann Whitney test p=2.1e-05), while the second PC (PC2; 13.6% of total variance explained) captured strain effects on the proteomic footprint only among EVs one tailed Mann Whitney t test for PC2 differences between *virR^mut^* EVs and those from either WT or *virR^mut^* -Comp strains (one-tailed Mann Whitney p=1.2 e-02). These results suggest that differences in protein abundances between EVs and WCLs are larger than differences found between strains, and that, concerning the differences across strains, these are more important among EVs. To verify this, we conducted differential protein expression analyses to find barely 27 proteins differentially expressed between *virR^mut^* and any of the two *virR*-expressing strains, which represents 2.9% of the total of proteins under analyses (*Supplementary file 4*). Despite the scarcity of significant differences in expression across strains in WCLs, some of the features most differentially expressed in *virR^mut^* are, besides VirR itself (*Figure 6C*), proteins involved in key processes for bacterial survival and virulence, including key metabolic enzymes such as the Acetyl-coenzyme A synthetase Acs (logFC = –3.92 (FDR = 4.5e-3) and –2.96 (FDR = 5.2E-2) for the differences between *virR^mut^* and WT, and between *virR^mut^* and *virR^mut^*-Comp, respectively), as well as FprA, (logFC = –3.12, FDR = 4.5e-3 e-02 for *virR^mut^*-WT and logFC = –2.16, FDR = 5.9 e-02 for *virR^mut^*-*virR^mut^*-Comp), involved in the reduction of NADP +into NADPH, both downregulated in *virR^mut^*.

Conversely to the WCL dataset, we did find more frequent differences in protein expression across strains in EVs, with as many as 361, and 382 proteins differentially expressed in *virR^mut^* EVs when compared to those from WT and *virR^mut^*-Comp strains, respectively (*Figure 6D*, *Supplementary file 4*). This represents 38.6 and 40.9% of all proteins tested, respectively. Similarly to what was observed at the level of RNA expression, differences in expression for the contrasts (*virR^mut^*-WT) and (*virR^mut^*-*virR^mut^*-Comp) were directionally enriched (*Figure 6E*), thus motivating the definition of two sets of proteins that (1) were observed at higher levels in the EVs isolated from *virR^mut^* than in those from both WT and *virR^mut^* -Comp strains (N=117 *virR^mut^* -upregulated proteins); or (2) were observed at lower levels in the EVs from *virR^mut^* than in those from any of the other two strains (N=108 *virR^mut^*-downregulated proteins). Interestingly, as many as 21 out of the 117 proteins that appeared enriched in *VirR^mut^* EVs were also upregulated in *virR^mut^* at the transcriptional level (One-sided Fisher exact test of enrichment OR = 2.02, p=7.0e-3) suggesting that at least a part of these differences in EVs' protein cargo are a consequence of differences in gene expression already observed at a transcriptional level. Notwithstanding that, proteins appearing at lower concentrations in *virR^mut^* EVs were not enriched among genes bearing significant transcriptional effects in the mutant.

Regarding downregulated EV proteins in *virR^mut^* we observed robust enrichments in GO terms related with protein biosynthesis, including translation (FDR = 9.8 e-22), ribosome (FDR = 5.3 e-29), and ribosome biogenesis (FDR = 4.1 e-27, *Figure 6F*, *Supplementary file 5*). Consistent with these results, 44 ribosomal proteins (out of 49 detected) were found to be significantly less expressed in the *virR^mut^* strain relative to either WT or *virR^mut^*-Comp strains (FDR < 0.05 for at least one comparison), while no ribosomal protein appeared significantly upregulated in *virR^mut^*. Furthermore, 30 of these proteins were simultaneously downregulated in *virR^mut^* when compared to both VirR-containing strains (*Figure 6—figure supplement 1*) implying a stark ribosomal depletion in the EVs in the absence of VirR. Furthermore, we found that the two bacterioferritin proteins BfrA and BfrB were simultaneously less abundant in *virR^mut^* than in both WT and *virR^mut^*-Comp strains (logFC <−1.5 and FDR < 7.3E-4, for differences *virR^mut^* vs WT and *virR^mut^* vs *virR^mut^*-Comp in both genes, *Figure 6G*, *Supplementary file 4*), together contributing to the enrichment of the GO term ferroxidase activity (FDR = 6.4 e-02, *Figure 6F*). This result is suggestive of a possible defect of *virR^mut^* EVs in iron sequestration (*Mohammadzadeh et al., 2021*; *Prados-Rosales et al., 2014b*). We also found a series of metabolic pathways enriched among proteins upregulated in the EVs of the *virR^mut^* strain, including enzymes involved in

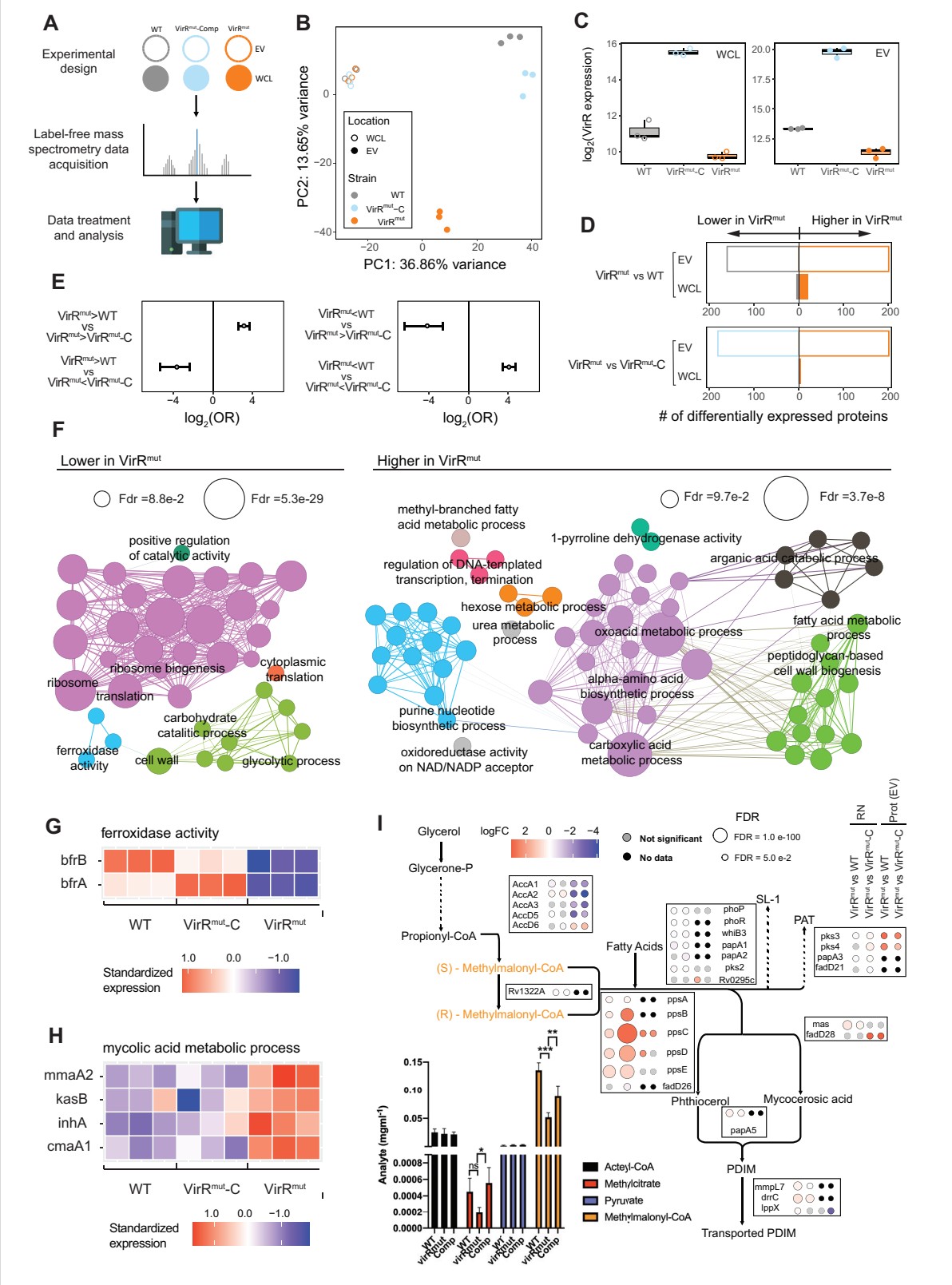

**Figure 6.** VirR regulates the protein content of EVs in *Mtb*. (**A**) Experiment design for proteomics analyses. label-free mass spectrometry data was retrieved for replicates of H37RV (WT), *virR^mut^* and *virR^mut^*-Comp. For each strain, data corresponding to proteins extracted from either whole-cell extracts (WC) and EVs was produced, totaling 12 samples (2 strains times 2 cell compartments times 3 replicates). (**B**) PCA plot of the proteomics data. The loadings from the two first principal components, adding up to 50.5% of variance explained, are plotted. (**C**) Expression patterns of VirR in the

*Figure 6 continued on next page*

*Figure 6 continued*

proteomics dataset across cell compartments and strains. As expected, while *virR^mut^* shows significantly lower levels of VirR protein, the complemented strain does not just restore WT levels of expression, but leads to a strong overexpression of VirR, both in the lysates and the EVs, (**D**) Bar plots of differentially expressed proteins found between strains in EVs (up: *virR^mut^* vs WT; down *virR^mut^* vs *virR^mut^*-Comp at 5% FDR). (**E**) Enrichment profiles between sets of proteins found up and down-regulated for the contrasts *virR^mut^* vs WT and *virR^mut^* vs *virR^mut^*-Comp. (**F**) Gene Ontology enrichments of the downregulated proteins in the VirR mutant strain (left) and the upregulated ones (Right), at the EV location. GO terms selected for testing included biological process and cell compartment labels containing N>3 genes, between levels 4–6 of the ontology tree. As in *Figure 3E*, terms selected for visualization with FDR = 10% and enrichment odds-ratio >2, nodes' size is proportional to the enrichment significance, and links between terms are added wherever the sets of proteins contributing to the enrichments in a connected pair shows an intersection larger than 50% of the smallest of the gene sets involved in the pair (see Materials and methods for further details, and *Supplementary file 5* for an extended list of all the enrichments found). (**G**) Expression patterns of bacterioferritin proteins in the EVs of WT and *virR^mut^*. (**H**) Expression patterns of proteins contributing to enrichments of terms related to mycolic acids biosynthesis. (**I**) Bottom left: Concentrations of key metabolites in the propionyl-coA detoxification routes. Right: Metabolic pathway of the propionyl-coA detoxification route through the methyl-malonyl route, along with the differential expression statistics between the WT and *virR^mut^* strains observed at transcriptomic and proteomic levels for the main enzymes involved. The framed plot represents the quantification of selected metabolites by mass spectrometry.

The online version of this article includes the following figure supplement(s) for figure 6:

**Figure supplement 1.** Expression patterns of ribosomal proteins in the EVs from WT, *virR^mut^* and *virR^mut^* -Comp strains.

**Figure supplement 2.** Expression patterns in the EVs from WT, *virR^mut^* and *virR^mut^* -Comp strains of proteins differentially expressed in *virR^mut^* that contribute to enrichments of terms related to metabolism of: (**A**) nucleotides (**B**) amino-acids (**C**) oligosaccharides (**D**) glycogen.

metabolism of nucleotides (e.g. purine nucleotide biosynthetic process FDR = 2.0e-02, amino acids (alpha amino acids FDR = 8.3e-04), as well as carbohydrates hexose metabolic process FDR = 6.3e-2), whose catabolism appears also enriched among proteins downregulated in the mutant (e.g. glycolytic process FDR = 2.2e-2 and carbohydrate catabolic process FDR = 4.7E-2). The simultaneous enrichments of these metabolic pathways observed both among upregulated and downregulated vesicular proteins (*Figure 6—figure supplement 2*), indicates a complex, VirR-dependent, rewiring of metabolic routes in the vesicles from the mutant strain.

Further, proteins upregulated in *virR^mut^* EVs (N=117) were enriched in terms related to mycolic acids metabolic processes (FDR = 2.1E-2), and peptidoglycan-based cell-wall biogenesis (FDR = 9.0 e-02). Proteins such as InhA (logFC = 3.71, 2.26 FDR = 2.20 e-05, and 1.4E-3 for comparisons *virR^mut^* vs WT and *virR^mut^* vs *virR^mut^*-Comp, respectively), and KasB (logFC = 1.83; 2.49 FDR = 2.23 e-02; 4.4e-3), involved in mycolic acid metabolism were also upregulated in *virR^mut^* (*Figure 6H*).

The upregulation of these proteins, along with the enrichments at the transcriptional level of DIM/DIP biosynthetic pathways (upregulated in *virR^mut^*), suggests a shift in the control of the pool of propionyl-CoA, both at the level of its production, -as a byproduct of the catabolism of methyl-branched lipids, odd-chain-length fatty acids as well as cholesterol- and its detoxification by the bacterium through different routes, such as the methylmalonyl pathway, the glyoxylate cycle and the methylcitrate cycles (*Lee et al., 2013*). To shed light on these changes at the metabolic level, we conducted a metabolic profiling to quantify the concentration of metabolites that are central in these different detoxification routes including propionyl-CoA: acetyl-CoA, pyruvate, methylcitrate, and methylmalonyl-CoA (*Figure 6I*). Only methylmalonyl-CoA was found to be significantly repressed in *virR^mut^* (p=0.028). We also observed differences between *virR^mut^* and *virR^mut^* -Comp strains in methylcitrate. This result, along with the trend towards increased protein, and especially gene expression that is observed for genes involved in the biosynthesis of PDIMs from methylmalonyl (including phenolpthiocerol synthesis type-I polyketide synthases A-E, (ppsA-E, *Figure 6I*), among others), suggests that the methylmalonyl pool is depleted in the mutant strain because of an increased activity of the PDIM biosynthetic pathway. Furthermore, most of the subunits of the acetyl/propionyl-CoA carboxylase Acc (AccA1, AccA2, AccA3, AccD1, AccD5) appear repressed in *virR^mut^* EVs (*Figure 6I*), suggesting that the production of methylmalonyl Co-A from the degradation of propionyl CoA is inhibited within EVs of the mutant strain.

Taken together, the omics analyses align with the structural and functional differences observed in *virR^mut^*. We reasoned that the increased expression of PDIMs biosynthetic genes and the reduced expression of genes related to protein synthesis may be triggered to compensate the enhanced permeability measured in *virR^mut^*. In addition, this enhanced permeability leads to an increase in the release of EV, which seems to be depleted in ribosomes and iron storage proteins.

## The absence of VirR leads to an aberrant enlargement of PG in *Mtb*

To elucidate the specific molecular defects that explain the periplasm enlargement, the enhanced permeability, and the antibiotic sensitivity phenotypes, we performed a biochemical analysis of WT and *virR^{mut}* cell envelopes. First, we analyzed the total lipid profile by liquid chromatography-mass spectrometry (LC-MS) and observed significant differences in prevalent polar lipids linked to the cell membrane including, lysophosphatidic and phosphatidic acid, species of monoacylated and diacylated glycerol, phosphatidylethanolamines and phosphatidyl-myo-inositol mannoside (PIM) species, being more abundant in *virR^{mut}* relative to WT and complementing strain (*Figure 7A*). To validate these results, we ran thin layer chromatography (TLC) experiments on the same lipid extracts and found that the mutant carries more phospholipids in the membrane including more AcPIM$_2$ and PE (*Figure 7B*). Moreover, we could not measure major differences in the level of mycolic acid species (*Figure 7C*), while significantly more PDIMs were detected in the mutant relative to the other strains (*Figure 7D*). This later result partially validates proteomics and metabolomic data and suggests that the methylmalonyl pool is depleted in the mutant strain because of an increased activity of the PDIM biosynthetic pathway. However, other lipids including SL1 or PAT/DAT that share a common precursor (propionyl- CoA) with PDIM may also explain the depletion of methylmalonyl. Notably, we observed a reduction in the level of these lipids (SL1 or PAT/DAT) in *virR^{mut}* cultured in glycerol relative to WT and complemented strains, suggesting that the excess of PDIM synthesis can take away methyl malonyl CoA from the biosynthesis of SL-1 and PAT/DAT in the absence of VirR (*Figure 7—figure supplement 1*).

When analyzing apolar lipids, we also observed increased levels of free mycolic acids (FMA) and free fatty acids (FFA) upon *virR* deletion (*Figure 7E*). The enhanced release of these apolar lipids has also been documented in a *C. glutamicum* mutant in Cg0847 (*Baumgart et al., 2016*), a canonical LCP protein. Importantly, we could not detect any of these lipids in EVs from *virR^{mut}* (*Figure 7F*), while major polar phospholipids could be resolved in EVs from all strains (*Figure 7G*), as previously reported (*Prados-Rosales et al., 2011*). This result indicates that *Mtb* releases free apolar lipids independently from membrane-associated EVs.

Next, we performed a glycosyl composition analysis of isolated cell walls (including mycolyl-arabinogalactan-peptidoglycan complex) and determined that there were no significant differences between strains in AG content as well as in the ratio between arabinose and galactose (A/G) (*Figure 8A*). Similarly, no changes were observed in the galactan length as measured by the ratio between galactose (G) and rhamnose (R) G/R (*Justen et al., 2020*). We noticed that the ratio between N-acetylglucosamine (NAGc) and rhamnose (R) (NAGc/R) was significantly higher in *virR^{mut}* relative to WT, suggesting an increase in PG content in this strain (*Figure 8A*). The validation of these results was performed independently by measuring the amount of diaminopimelic acid (DAP) in isolated cell walls, as a surrogate of PG content, and determined that *virR^{mut}* carries two times more PG than WT or complemented strains (*Figure 8B*).

We next obtained pure PG preparations of each strain and performed an analysis by atomic force microscopy (AFM; *Figure 8C* and *Figure 8—figure supplement 1*). This technique has been recently used to investigate the nanometric features of isolated Gram-positive bacterial cell walls (*Pasquina-Lemonche et al., 2020*). Semi-automated analysis of liquid AFM images from mycobacterial isolated PG samples revealed that: (i) the absence of VirR leads to a significant reduction in PG pore diameter (down to 9.2±0.6 nm) relative to WT (16±3.3 nm) and complemented strains (10±0.8 nm; *Figure 8C and D* and *Figure 8—figure supplement 1*); (ii) the number of pores is significantly higher in *virR^{mut}* compared to WT and complemented strain (*Figure 8E* and *Figure 8—figure supplement 1*); (iii) manual analysis of AFM images in ambient conditions revealed that *virR^{mut}* PG is significantly thicker than that of the other strains of the study (*Figure 8F*). These results show for the first time the nanometric architecture of isolated PG in mycobacteria by AFM and suggest a role for VirR in regulating PG thickness and porosity. Strikingly, we found that the PG enlargement observed in the *virR^{mut}* correlated with increased susceptibility to the muramidase lysozyme (*Figure 8—figure supplement 2*), indicating modifications in PG remodeling in *virR^{mut}*. In fact, a deeper MS analysis of PG showed a significant increase in the abundance of deacetylated muropeptides in *virR^{mut}* relative WT and *virR^{mut}*-Comp strain (*Figure 8G and H*). This PG feature has been linked to variability in lysozyme susceptibility in other Gram-positive bacterial species (*Bera et al., 2005*). Collectively, although the structural changes at nanoscopic level of the peripheral areas of the cell observed by AFM leads to a more dense and

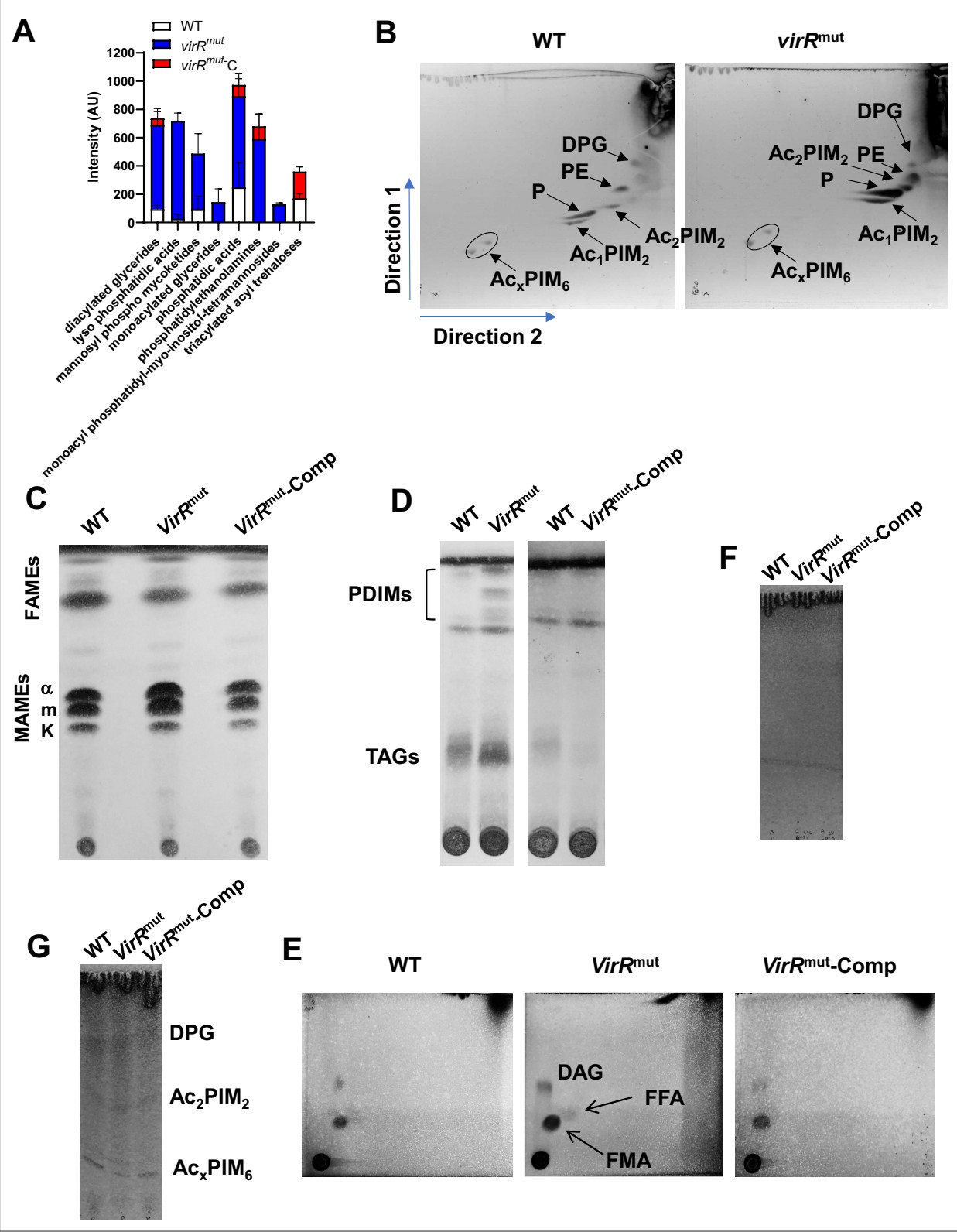

**Figure 7.** Lipidomic analysis of the cell envelope and EVs from *virR^mut^*. (**A**) Comparative total lipid analysis of the indicated strains by nano-LC-MS. Lipid species with an abundance higher than 0.5% are shown. Data indicates mean and standard error from three biological replicates. (**B**) Analysis of polar lipids of the indicated strains by bidimensional (2D)-layer chromatography (TLC) (2D-TLC). Phosphatidyl-myo-inositol dimannosides (Ac$_1$PIM$_2$ and Ac$_2$PIM$_2$, respectively); phosphatidyl-myo-inositol hexamannosides (Ac$_x$PIM$_6$); phosphatidylethanolamine (PE); diphosphatidylglycerol (DPG).

*Figure 7 continued on next page*

*Figure 7 continued*

Phospholipid (P). (**C**) Analysis of mycolic acid species in the indicated strains by TLC. (**D**) Analysis of PIDMs and TAGs by TLC. (**E**) Analysis of apolar lipids of the indicated strains by TLC. Diacylglycerol (DAG); free fatty acid (FF); free mycolic acid (FMA). (**F**) Analysis of apolar lipids of MEVs isolated from the indicated strains. Triacylglycerols (TAGs); phthiocerol dimycocerosates (PDIMs) (**G**) Analysis of polar lipids of MEVs isolated from the indicated strains. Phosphatidyl-myo-inositol dimannosides ($Ac_1PIM_2$ and $Ac_2PIM_2$, respectively); phosphatidyl-myo-inositol hexamannosides ($Ac_xPIM_6$); diphosphatidylglycerol (DPG).

The online version of this article includes the following source data and figure supplement(s) for figure 7:

**Source data 1.** Pngs containing original TLC images, indicating the relevant elements displayed in *Figure 7*.

**Source data 2.** Original files of 1D & 2D TLC images displayed in *Figure 7*.

**Figure supplement 1.** The absence of VirR reduces the membrane abundance of sulfolipids.

**Figure supplement 1—source data 1.** Png containing original 2D-TLC images, indicating the relevant elements displayed in *Figure 7—figure supplement 1*.

**Figure supplement 1—source data 2.** Original files for the TLC assays displayed in *Figure 7—figure supplement 1*.

thick structure harder to penetrate, the chemical changes in the peptidoglycan structure observed and biochemical assays suggest that changes associated with alterations in VirR-regulated PG remodeling makes the wall more permeable and support increased release of EV in *Mtb*.

## VirR interacts with canonical LCP proteins

As mentioned above VirR is a LytR_C solo domain protein with no catalytic LCP domain. Other members of the LCP protein family like Rv3267 and Rv3287 are the main PG-AG ligases (*Grzegorzewicz et al., 2016*; *Harrison et al., 2016*) and also carry a LytR_C domain. We have previously shown that VirR interacts with itself (*Rath et al., 2013*). Consequently, we reasoned that if VirR contributed to PG-AG ligation it could do it via interaction with canonical LCP proteins through the LytR_C domain. Although VirR is one of the most abundant proteins found in tuberculin preparations (*Prasad et al., 2013*), we determined that it localizes at sites of nascent cell wall by fluorescence microscopy (*Figure 9—figure supplement 1*). Therefore, its location could initially allow for interactions with other cell surface proteins like LCP. Both recombinant VirR and the main AG-PG ligase Lcp1 (Rv3267) oligomerize in a blue-native PAGE gel (*Figure 9A*). This observation suggests that they interact with proteins sharing similar domain features. We next performed a flotation assay where we examined the behavior of VirR on an iodixanol gradient in the presence or absence of Lcp1. Using a specific antibody raised against VirR, we observed that the presence of Lcp1 displaces the relative location of VirR toward upper fractions of the gradient, indicating a potential interaction of both proteins and the formation of higher molecular weight complexes (*Figure 9B*). To investigate the interaction of virR and Lcp1, along with other Lcp proteins, in a more physiological environment we used the bipartite splitGFP approach (*Cabantous et al., 2013*; *Figure 9C* and *Figure 9—figure supplement 2*), which has been recently utilized to demonstrate the co-localization and interaction of the membrane-bound metal ATPase CtpC with the chaperon PacL1 (*Boudehen et al., 2022*). We first recapitulated the VirR self-interaction and added that such interaction implicates the LytR_C domain since no differences in fluorescence were measured between full and truncated versions of VirR (VirRsol) lacking the first 41 amino acids. As negative controls, we included interactions between VirR and the cell surface associated metal ATPase CtpC (*Figure 9C*). When testing interactions between VirR and other Lcp proteins (Rv0822c, Lcp1 (Rv3267) and Rv3284) we could not measure any GFP signal when full Lcp proteins were used in the assay. Conversely, a significant increase in fluorescence was measured for all combinations when truncated Lcp proteins including either the C-terminal domain or specifically the LytR_C domain were tested (*Figure 9C*). These results strongly suggest that VirR interacts with Lcp proteins and that these interactions occur via LytR_C terminal domains.

## Discussion

We reported *virR* as the first gene implicated in the biogenesis of EVs in *Mtb* 10 years back (*Rath et al., 2013*). A transposon mutant in *virR* (*virR^mut^*) overproduces MEVs and this provokes augmentation of cytokine responses in mouse and human macrophages. This mutant manifested an attenuated phenotype in experimental infections on macrophages and mice (*Beaulieu et al., 2010*; *Rath et al.,*

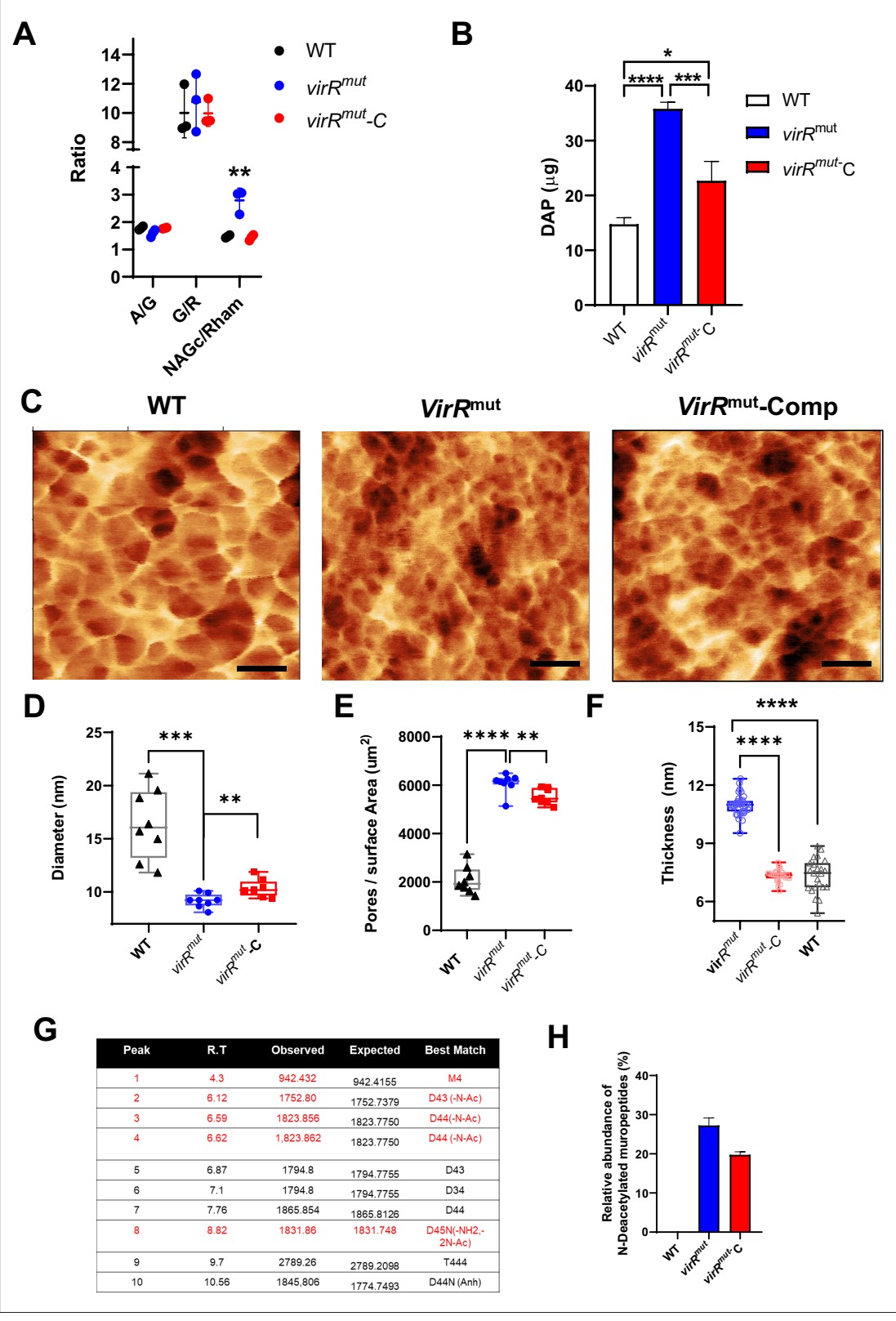

**Figure 8.** Chemical and nanometric analysis of isolated cell walls shows higher amounts of PG in the absence of VirR. (**A**) Glycosyl composition analysis of isolated cell walls from the indicated strains. Acid hydrolysis prior to GC-MS analysis of isolated cell walls was performed to determine the total amounts of galactose (G), arabinose (A), rhamnose (R), and N-acetylglucosamine (NAGc). The ratio between different sugars is shown to indicate: (i) the relative levels of arabinogalactan (A/G); (ii) the length of the AG polysaccharide chain (G/R); (iii) the amount of PG (NAGc/Rham). Individual

*Figure 8 continued on next page*

*Figure 8 continued*

measurements are shown from three biological replicates. Data are mean and standard errors. \*\*p<0.01 after applying a Tukey´s multiple comparison test. (**B**) Quantification of diaminopimelic acid (DAP) on isolated cell walls from the indicated strains. Data are mean and standard errors from three biological replicates; \*\*\*\* denotes p<0.0001, \* denotes p=0.0119 after applying a Tukey's multiple comparison test. (**C**) Atomic force microscopy high-resolution images taken in liquid from the external surface of purified peptidoglycan from different samples, left to right: WT, *virR^{mut}*, *virR^{mut}-C*, the structure of external peptidoglycan layer shows a mesh with pores of different sizes, (**D**) Graph showing the pore diameter from n=8 AFM images similar to (**C**) (each point represent an image) of different samples; black triangles: WT, blue circles: *virR^{mut}*, red squares: *virR^{mut}-C*. The pore diameter was calculated from the binary slice from each image containing the greatest number of pores. (**E**) The number of pores per surface area from different samples; white triangles: WT, red circles: *virR^{mut}*, blue squares: *virR^{mut}-C*, this was calculated from the slice containing the maximum number of pores where the pore size was analyzed. (**F**) Graph showing the peptidoglycan thickness manually measured using the 1D statistic tools from the open-source program Gwyddion (**Nečas and Klapetek, 2012**) from AFM images in air containing several fragments of cell wall, each point represents a different peptidoglycan fragment from a different cell. There were three samples analyzed: black WT n=23 peptidoglycan fragments, blue *virR^{mut}* n=36 and red *virR^{mut}-C* n=25. Statistics: (**D**) Using a two-tailed t test with Welch's correction the statistical comparison are: (\*\*\*) pWT- *virRm^{ut}* = 4.1 10^{-4} with t=6.1, df = 7.5, (\*\*) p *virR^{mut} - virRm^{ut-C}*-C=0.007 with t=3.2, df = 13.2, (**E**) Using a two-tailed t test with Welch's correction the statistical comparison are: (\*\*\*\*) pWT- *virRm^{ut}* = 6.6 10–10 with t=16.3, df = 12.7, (\*\*) p *virR^{mut} - virR^{mut}*-C=0.01 with t=3.0, df = 13.2., (**F**) Using a two-tailed t test with Welch's correction the statistical comparison are: (\*\*\*\*) pWT- *virRm^{ut}* = 2.8 10–18 with t=17.8, df = 32.3, (\*\*\*\*) *virR^{mut} - virR^{mut}*-C=3.9 10–17 with t=16.4, df = 31.8. (**G**) List of muropeptides identified in the cell walls of Mtb. Features including retention time (RT), observed and expected mass as well as the best match for each peak are shown. Those deacetylated muropeptides are indicated in red. (**H**) Relative abundance of deacetylated muropeptides in the indicated strains.

The online version of this article includes the following figure supplement(s) for figure 8:

**Figure supplement 1.** Additional AFM data.

**Figure supplement 2.** Sensitivity to lysozyme of *virR^{mut}*.

*2013*). Such studies established that VirR has a role in virulence by negatively modulating the release of MEVs. These studies provided evidence that vesiculogenesis in *Mtb* is genetically regulated. Since then, conditions such as iron starvation (**Prados-Rosales et al., 2014b**) and other genes, including the bacterial dynamin-like *iniAC* (**Gupta et al., 2023**), and the Pst/SenX3-RegX3 signal transduction system (**White et al., 2018**) have been shown to contribute to the biogenesis of MEVs. Interestingly, VirR, a LytR_C solo domain protein (**Figure 1—figure supplement 1**), shares homology with Lcp proteins, which in *Mtb* are known to participate in the enzymatic connection between AG and PG (**Grzegorzewicz et al., 2016**; **Harrison et al., 2016**). This suggests VirR function might be linked to cell wall remodeling.

In this study, we investigated how VirR can control the magnitude of MEV release in *Mtb*. We did this by studying the underlying enhanced vesiculogenesis phenomenon observed in *virR^{mut}* using genetic, transcriptiomics, proteomics and ultrastructural and biochemical methods. It was clear from our cryo-EM studies that *virR^{mut}* has an aberrant cell wall as the layer above the cell membrane was significantly larger than that of WT and complemented strains. The enlarged compartment showed higher granular texture, a feature that has been previously associated to PG precursors in unrelated Gram-positive bacteria (**Zuber et al., 2006**). This observation and previous studies showing the increased sensitivity of *VirR^{mut}* to vancomycin (**Ballister et al., 2019**) strongly suggested that the cell wall defect is connected to PG remodeling. The use of AFM on isolated PG confirmed this and showed for the first time the nanostructure of PG in *Mtb*, providing information on PG thickness, pore size and number of pores. AFM indicated that *virR^{mut}* PG is thicker, has significantly more pores of small size. Considering the pore sizes measured across strains (ranging from 9 to 16 nm), we initially ruled out the notion that EVs (which are spheres of 50–300 nm in diameter) are released through the peripheral cell wall. Therefore, *Mtb* must have an alternative mechanism to release EVs, possible during the division process where the PG is most fragile, presumably due to defects in crosslinking, something that awaits further experimental verification. These events were not possible to visualize in AFM given that PG fragments from all the strains were lacking the poles or clearly defined division sites but could be the subject of further studies in the future. However, we cannot exclude that MEV plasticity could allow for crossing the cell wall via PG pores.

A recent study showed that *Corynebacterium glutamicum* can produce and release different types of MVs through different routes, including mycomembrane blebbing or bubbling cell death, depending on the stress at which *C. glutamicum* is exposed (**Nagakubo et al., 2021**). In the present study, we have shown that *virR^{mut}* overproduces EVs under normal growing conditions in the absence of cell lysis, according to our assay which measured cytoplasmic leaking of the transcription factor IdeR or serially diluted spots on solid medium. Overall, we show that EV production in *Mtb* under our

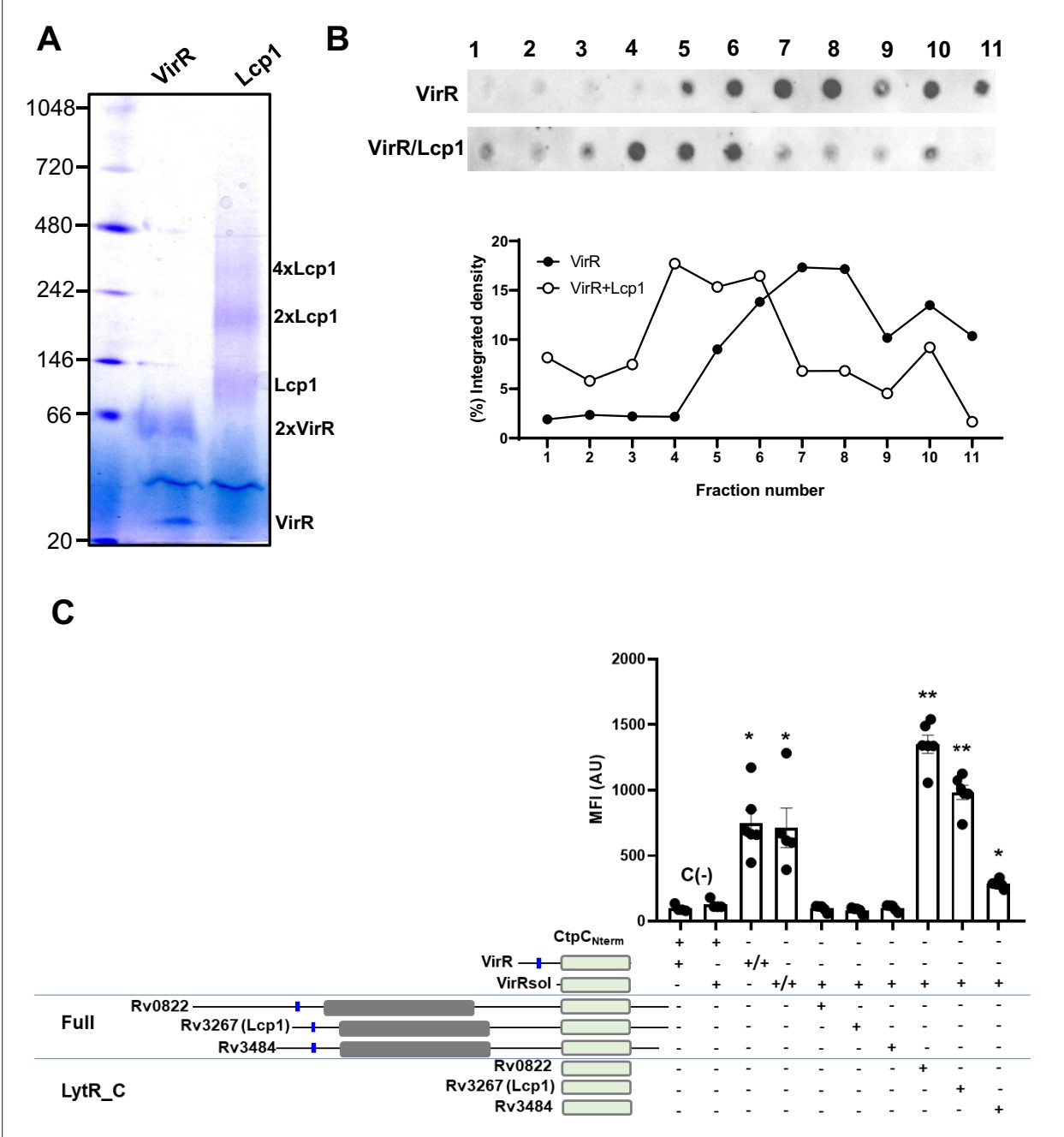

**Figure 9.** VirR interacts with Lcp1. (**A**) Blue-native PAGE analysis of recombinant VirR and Lcp1. The molecular mass of markers in kDa are indicated on the left side of the gel. The size of the multimers of VirR and Lcp1 are indicated on the left side of the gel. (**B**) *Upper panel*. Representative image of a flotation assay of recombinant VirR alone or in combination with Lcp1 performed on an idoxanol density gradient. The presence of VirR was detected in each fraction by dot-blot using murine polyclonal antibodies raised against VirR. Numbers indicate collected fractions from top (1) to bottom (11). *Lower panel*. Quantification of the dot-blot by measuring the pixel intensity of the dots using ImageJ. Data are mean and standard error of three independent experiments. (**C**) Bipartite split-GFP experiment using *M. smegmatis* D2 expressing plasmids depicted in (*Figure 9—figure supplement 2A*), including CtpC_Nterm, VirR, VirRsol (Δ1–41), Rv3484, Rv3267 (Lcp1), Rv0822, Rv3484_LytR_C, Rv3267_LytR_C and Rv0822_LytR_C. Bacteria were grown in complete 7H9 and fluorescence was recorded by FACS. Pooled data obtained from four to six independent cultures of each strain are shown. Statistical analysis was performed using non-parametric two-sided Mann-Whitney test. ns: not significant. * and ** refer to p-values <0.05 or 0.01, respectively.

The online version of this article includes the following source data and figure supplement(s) for figure 9:

**Source data 1.** Pngs containing original Blue-native PAGE analysis gel images, indicating the relevant elements displayed in *Figure 9*.

**Source data 2.** Original files of Blue-native PAGE analysis gel images displayed in *Figure 9*.

*Figure 9 continued on next page*

**Figure supplement 1.** Localization of VirR by fluorescence microscopy.

**Figure supplement 2.** Split-GFP strategy.

growing conditions, is not linked to cell death. This fact largely excludes the bubbling cell death mechanism. Moreover, our past and current lipidomic analysis (*Prados-Rosales et al., 2011*) indicates the absence of mycomembrane lipids in isolated MEVs from all strains, ruling out that MEVs originate at the mycomembrane. These apparent disagreements with the *C. glutamicum* study (*Nagakubo et al., 2021*) might indicate that *Mtb* EV biogenesis may differ from that of *C glutamicum*, and other mycolic acid-containing bacteria with shorter fatty acids. Additional variables that may explain differences between studies is the growth media and the time at which vesicles are isolated (*Gupta et al., 2023*). Of note, *M. smegmatis* EVs were isolated at day 6 (*Nagakubo et al., 2021*), a time at which this fast-growing mycobacteria is already in stationary phase.

The transcriptional profile of *virR^mut^* can provide a partial explanation for the attenuated phenotype of the mutant due to a downregulation of genes activated upon stresses associated to the intraphagosomal environment. It is particularly interesting to observe the specific reduction in transcript levels of the two most important effectors of the esx-1 secretion system. On the other hand, genes involved in PDIM synthesis were upregulated. It is possible that the specific upregulation of PDIM synthesis in absence of *virR* is part of compensatory response to a reduced permeability. We were surprised to find no differences between protein profiles in whole cells between WT and *virR^mut^* strains, despite difference in transcriptomics. A biological interpretation of these results would point to the existence of global compensatory mechanisms of post-transcriptional regulation used by the bacteria to partially recover comparable proteomic profiles in *virR^mut^* to those observed in the WT.

Contrary to whole cell lysates, EV protein profiles between *virR^mut^* and WT were largely different. In this context, it is accepted that certain proteins and lipids are enriched in bacterial EVs and that preferential inclusion of such antigens in MVs supports a specific vesicle biogenesis mechanism. In Gram-positive bacteria and mycobacteria both membrane and cytosolic proteins are included in EVs since the origin of these structures is linked to the cell membrane (*Prados-Rosales et al., 2011*) and, presumably, the incorporation of intracellular proteins may occur during the process of vesicle formation. We found a higher abundance of proteins related to PG and MA synthesis in *virR^mut^* EVs relative to WT. Therefore, these two processes seem to contribute to EV biogenesis in *Mtb*. Additional data presented in this study, which includes ultrastructural and chemical analyses confirm this. Both PG and MA synthesis are complex processes, including a diverse set of genes. What parts of the biogenesis of these molecules are important for the formation of EVs is a matter of future investigations.

Another finding of the study is the connection between permeability and the magnitude of the vesiculation process in *Mtb*. We linked these two phenomena first, by measuring enhanced vesiculation and permeability in the same strain (*virR^mut^*) and second, by modifying the permeability using chemical (via sublethal exposure to TRZ) and nutritional supplements (via providing cholesterol as sole carbon source) in WT *Mtb*. TRZ, a repurposed drug that belongs to the class of phenothiazines, is known to affect respiration in *Mtb* (*Amaral et al., 1996*). It is also assumed that TRZ can inhibit efflux pumps leading to an increased susceptibility to first-line antitubercular drugs when tested in combination (*de Knegt et al., 2014*). This provides an explanation for the synergistic effects of such combinations. An independent explanation of such synergy may also come from the increased permeability, as it has been recently shown by a quantitative proteomics analysis of *Mtb* cells exposed to sublethal concentration of TRZ (*de Keijzer et al., 2016*). When we grew *Mtb* with cholesterol as a sole carbon source, a condition that slows mycobacterial growth but also reduces cell permeability via cholesterol accumulation in the cell wall (*Brzostek et al., 2009*), we observed the opposite trend on MEV biogenesis. We consider this finding of high relevance since *Mtb* permeability is one the major concerns when developing potential antimycobacterial drugs. Nevertheless, bacterial cell permeability is a complex property that it is controlled by physiochemical, biological, and chemical processes such as stereochemistry, lipophilicity, saturation and unsaturation, flexibility, viscosity, fluidity, pressure, temperature and physiological conditions (*Nagamani and Sastry, 2021*). How these variables change to allow the release of MEVs is not known.

This study also proposes a model VirR function whereby VirR may work as a scaffold for canonical Lcp proteins, which in *Mtb* links PG to AG. Based on the demonstration that VirR interacts with

canonical Lcp proteins, we believe that the absence of VirR deregulates the localization of canonical Lcp proteins. Although we cannot rule out the contribution of other phenomena including alteration of the PG molecule, VirR is a central scaffold in the cell wall remodeling process. Taking into consideration the fact that *VirR^{mut}* is attenuated in virulence on preclinical models of infections and manifests an enhanced vesiculogenesis and permeability, VirR may represent a novel and interesting drug target. We envision that targeting VirR may not only make *Mtb* weaker and more permeable to other antitubercular drugs, but to stimulate the immune system, via MEVs, to fight infection.

## Materials and methods

### Bacterial strains, media and growth conditions

*Mtb* H37Rv (ATCC) and derivative strains *virR^{mut}* and *virR^{mut}-C* (complemented strain; *Rath et al., 2013*) were used in this study. Bacteria were initially grown in Middlebrook 7H9 medium (7H9) supplemented with 10% (v/v) oleic acid-albumin-dextrose-catalase (OADC) enrichment (Becton Dickinson Microbiology Systems, Spark, MD, United States), 0.5% (v/v) glycerol and with Tyloxapol 0.05% (v/v; Sigma-Aldrich, Burlington, MA, United States) prior to inoculation in a minimal medium (MM) consisting of $KH_2PO_4$ 1 g/l, $Na_2HPO_4$ 2.5 g/l, asparagine 0.5 g/l, ferric ammonium citrate 50 mg/l, $CaCl_2$ 0.5 mg/l, $ZnSO_4$ 0.1 mg/l, without Tyloxapol 0.05% (v/v), containing 0.7% (v/v) glycerol, pH 7.0.

*Mtb* was also cultured in MM in the presence of sublethal concentrations of thirodazine (TRZ) of 6 µgml$^{-1}$ as previously described. We determined that this concentration does not affect cell growth. *Mtb* was also cultured in MM with cholesterol as a sole carbon source as previously described (*Brzostek et al., 2009*).

*Mtb* expressing fluorescent VirR was generated by cloning VirR into pMV261-Venus using fast cloning approach (*Li et al., 2011*).

*M. smegmatis* mc²155 (ATCC 700084) and recombinants derived from this strain were grown at 37 °C in 7H9 medium (Difco) supplemented with 10% albumin-dextrose-catalase (ADC, Difco) and 0.05% Tween-80 (Sigma-Aldrich), or on complete 7H11 solid medium (Difco) supplemented with 10% OADC (Difco). When needed, streptomycin (25 µg ml$^{-1}$) was added to the culture media. *Escherichia coli* strains StellarTM (Takara bio, San Jose, CA, United States) were grown at 37 °C in LB broth (Difco) or on L-agar plates (Difco) supplemented with streptomycin (25 µg ml$^{-1}$) when required.

### *M. smegmatis* expression vectors for split GFP experiment

Plasmids used for split GFP experiments were constructed by modifying the plasmid pGMCS-PpacL1-pacL1-Lk2-GFP11-ctpC(1-85)-Lk15-GFP(1-10) (*Boudehen et al., 2022*; *Supplementary file 6*). Plasmids pGMCS, conferring resistance to streptomycin, are shuttle vectors, episomal in *E. coli* and integrative in mycobacteria through insertion at the attL5 mycobacteriophage insertion site in the glyV tRNA gene. Lk2 (LEGSGGGGSGGGS) and Lk15 (GPGLSGLGGGGGSLG) are flexible linkers separating the protein domains of interest from the last 11th b-sheet or the first ten b-sheets of GFP, respectively. First, the Zinc-inducible PpacL1 promoter was replaced by P1, a strong constitutive promoter from *M. smegmatis* (*Ariyachaokun et al., 2020*), resulting in pGMCS-P1-pacL1-Lk2-GFP11-ctpC(1-85)-Lk15-GFP(1-10). Second, the pacL1 (Rv3269) sequence was replaced by that of virR (encoding amino-acids 1–164) or virRsol (encoding amino-acids 42–164) by In-Fusion HD cloning (Takara). PCR fragments were amplified using appropriate primer pairs (*Supplementary file 6*) and Phusion High-Fidelity PCR Master Mix with GC Buffer (Thermo Scientific, Walthman, MA, United States). Linear fragments were purified on agarose gels and inserted into pGMCS backbone linearized by PCR amplification with appropriate primers, using the In-Fusion HD Cloning Kit (Takara), following the manufacturer's instructions. Finally, the ctpC sequence was replaced on the resulting plasmids by the virR or virRsol sequences, or that encoding either the C-terminal part or only the LytR_C domains of the Rv0822c, Rv3267 or Rv3484 proteins from *M. tuberculosis* H37Rv. Plasmids were constructed by transformation into *E. coli* Stellar recipient cells. All plasmids were verified by sequencing before introduction by electroporation into *M. smegmatis* mc²155 strain.

### Peptidoglycan labeling with fluorescence D-amino acids (FDAAs)

For sequential labelling *M. tuberculosis* was initially inoculated at an OD$_{600}$ of 0.1 into 7H9 medium supplemented with 1 mM HADA and incubated for 48 hr. Bacteria were washed three times with 7H9

medium and incubated in 7H9 medium supplemented with the second FDAA (FDL) for 5 hr. To visualize FDAAs incorporation, bacterial suspensions were washed three times with PBS-0.05% Tyloxapol and fixed with 4% paraformaldehyde for 6 hr. Importantly, no growth defects were observed upon labeling with FDAAs. Fixed bacterial suspensions were mounted on agar pads and imaged on a fluorescence microscope using a 100 x oil immersion objective on an Orca 12 ERG digital CCD camera (Hamamatsu Photonics) mounted on a Nikon E600 microscope.

## $H_2O_2$ sensitivity assays

Bacterial strains were initially cultured in 7H9+ADN until they reached an OD600nm of 0.5. Cells were washed in PBS and transferred to roller bottles including the same medium and 13 mM $H_2O_2$. After 8 hr, bacterial cultures were serially diluted and spotted onto 7H10 solid medium.

## Generation of the CRISPR mutants

To generate the conditional CRISPR mutants we took advantage of the CRISPR interference (CRISPRi) technology (**Rock et al., 2017**). Briefly, we designed sgRNA oligonucleotides containing the target Rv0431 (*virR*) and Rv2700 (*cei*) genes sequences (**Supplementary file 6**) located immediately upstream of the PAM sequence. The strengths of the PAM sequence were chosen following authors recommendation to get high and intermediate gene repression, 158.1 or 42.2-fold repression for *virR* and 145.2 and 47.3 for *cei*, respectively. The primers were annealed and ligated into the gel purified BsmBI-digested CRISPRi vector (plJR965 plasmid) followed by transformation of competent *Mtb* cells via electroporation. Transcriptional knockdown of genes were quantified by qRT-PCR; and protein levels were examined by immunoblot using specific antibodies raised against VirR (**Rath et al., 2013**) and Cei (**Ballister et al., 2019**). Complementation of *virR* mutant strain was performed using the plasmid previously generated by Gateway Cloning Technology (Invitrogen) (**Rath et al., 2013**). Importantly, neither *virR*^mut nor the *virR*^mut-C showed altered expression of downstream genes including Rv0432 or Rv0433 relative to WT strain (**Figure 2—figure supplement 2**).

## EV isolation and purification

*Mtb*-EV were prepared and purified as previously described (**Prados-Rosales et al., 2011**). Briefly, the culture filtrate of 1 L cultures of *Mtb* grown in low iron MM without detergent for 14 days was processed for vesicle isolation by differential centrifugation. In parallel, a 2 ml culture in MM supplemented with 0.05% Tyloxapol to disperse bacterial clumps was used to determine viability by plating culture dilutions onto 7H10 agar plates and enumerating colony forming unit (CFU) at the time of culture filtrate collection.

The membranous pellet containing vesicles obtained after ultracentrifugation of the culture filtrate at 100,000× *g* for 2 hr at 4 °C, was resuspended in 1 ml sterile phosphate-buffered saline (PBS) and overlaid with a series of Optiprep (Sigma-Aldrich) gradient layers with concentrations ranging from 30% to 5% (w/v). The gradients were centrifuged at 100,000× *g* for 16 hr. At the end 1 ml fractions were removed from the top, diluted to 20 ml with PBS and purified vesicles recovered by sedimentation at 100,000× *g* for 1 hr. Vesicle pellets were suspended in PBS before analysis. Protein concentration was measured by Bradford assay (Bio Rad, Hercules, CA, United States) according to manufacturer instructions. For all conditions test from which EVs were isolated, normalization was performed by referring EVs concentration to colony forming units (CFUs).

## Nano particle tracking analysis (NTA)

NTA was conducted using ZetaView (Particle Metrix, Winning am Ammersee, Germany). Instrument calibration was performed prior to EV analysis using 102 nm polystyrene beads (Thermo Fisher Scientific), according to manufacturer instructions. Measurements were performed using a 405 nm 68 mW laser and CMOS camera by scanning 11 cell positions and capturing 60 frames per position at 25 C with camera sensitivity 85, shutter speed 100, autofocus and automatic scattering intensity. Samples were diluted in pre-filtered PBS to approximately 106–107 particles·ml⁻¹ in Millipore DI water. Analysis was performed using ZetaView Software version 8.05.12 SP1 with a minimum brightness of 30, maximum brightness of 255, minimum area of 5, maximum area of 1000, and s minimum trace length 15. Triplicate videos of each sample were taken in light scatter mode. Particle size and concentration

were analyzed using the built-in EMV protocol and plotted using Prism software, version 8.0.1 (GraphPad Inc, San Diego, CA, United States).

## Electron microscopy

Cells were fixed with 2% glutaraldehyde in 0.1 M cacodylate at room temperature for 24 hr, and then incubated overnight in 4% formaldehyde, 1% glutaraldehyde, and 0.1% PBS. For scanning microscopy, samples were then dehydrated through a graded series of ethanol solutions before critical-point drying using liquid carbon dioxide in a Toumisis Samdri 795 device and sputter-coating with gold-palladium in a Denton Vacuum Desk-2 device. Samples were examined in a Zeiss Supra Field Emission Scanning Electron Microscope (Carl Zeiss Microscopy, LLC, North America), using an accelerating voltage of 5 KV.

For Cryo-EM, grids were prepared following standard procedures and observed at liquid nitrogen temperatures in a JEM-2200FS/CR transmission electron microscope (JEOL Europe, Croissy-sur-Seine, France) operated at 200 kV. An in-column omega energy filter helped to record images with improved signal/noise ratio by zero-loss filtering. The energy selecting slit width was set at 9 eV. Digital images were recorded on an UltraScan4000 CCD camera under low-dose conditions at a magnification of 55,058 obtaining a final pixel size of 2.7 Å/pixel.

Density profiles were calculated along rectangular selections with the ImageJ software (NIH, Bethesda, MD, United States). The number of cells analyzed varied from n=20 (WT), n=15 ($virR^{mut}$), and n=23 ($virR^{mut}$-C). The comparisons extracted from this data is statistically relevant with t test with Welch's correction performed. For each cell three representative measurements were obtained. Mean and standard error was then calculated for measurements pooled for each strain.

## Western dot blot

Two µl of EV isolates and twofold serial dilutions were spotted onto a nitrocellulose membrane (Abcam, Cambridge, United Kingdom) and process for dot blot using anti-EV polyclonal murine serum (**Prados-Rosales et al., 2014a**) as primary antibody and goat anti-mouse-HRP conjugated as secondary antibody and ECL prime Western Blotting Detection chemiluminescent substrate (GE Healthcare, Chicago, IL, United States). The signal was visualized in a Chemidoc MP imaging system (Bio-Rad)s.

## RNA sequencing

*Mtb* strains were grown in MM supplemented with 0.05% Tyloxapol at 37 °C until they reached an OD595nm of 0.3 and harvested by centrifugation. The cell pellets were resuspended in 1 ml Qiagen RNA protect reagent (QIAGEN, Venlo, Netherlands) and incubated for 24 hr at room temperature. Cells were disrupted by mechanical lysis in a FastPrep-24 instrument (MP Biomedicals, Santa Ana, CA, United States) in Lysing Matrix B tubes and RNA was purified with the Direct-zol RNA miniprep kit (Zymo Research, Irvine, CA, United States). The quantity and quality of the RNAs were evaluated using Qubit RNA HS Assay Kit (Thermo Fisher Scientific) and Agilent RNA 6000 Nano Chips (Agilent Technologies, Santa Clara, CA, United States), respectively. Sequencing libraries were prepared using the Ribo-Zero rRNA Removal Kit (Gram-positive Bacteria; Illumina Inc, San Diego, CA, United Stated) and the TruSeqStranded mRNA library prep kit (Illumina Inc), following the Ribo-Zero rRNA Removal kit Reference guide and the "TruSeqStranded mRNA Sample Preparation Guide. Briefly, starting from 1 µg of total RNA, bacterial rRNA was removed and the remaining RNA was cleaned up using Agencourt RNAClean XP beads (Beckman Coulter). Purified RNA was fragmented and primed for cDNA synthesis. cDNA first strand was synthesized with SuperScript-II Reverse Transcriptase (Thermo Fisher Scientific) for 10 min at 25 °C, 15 min at 42 °C, 15 min at 70 °C and pause at 4 °C. cDNA second strand was synthesized with Illumina reagents at 16 °C for 1 hr. Then, A-tailing and adaptor ligation were performed. Libraries enrichment was achieved by PCR (30 s at 98 °C; 15 cycles of 10 s at 98 °C, 30 s at 60 °C, 30 s at 72 °C; 5 min at 72 °C and pause at 4 °C). Afterwards, libraries were visualized on an Agilent 2100 Bioanalyzer using Agilent High Sensitivity DNA kit (Agilent Technologies) and quantified using Qubit dsDNA HS DNA Kit (Thermo Fisher Scientific). Library sequencing was carried out on an Illumina HiSeq2500 sequencer with 50 nucleotides single end reads and at least 20 million reads per individual library were obtained.

## RNA-sequencing data analysis

Quality Control of sequenced samples was performed by FASTQC software (http://www.bioinformatics.babraham.ac.uk/projects/fastqc/). Reads were mapped against the *M. tuberculosis* H37Rv strain (GCF_000195955.2_ASM19595v2) reference genome using Tophat (*Trapnell et al., 2009*) with `--bowtie1` option, to align 50 bp reads. The resulting BAM alignment files for the samples were the input to Rsubread's (*Liao et al., 2019*) featureCounts function to generate a table of raw counts required for the Differential Expression (DE) analysis. Data preprocessing and differential expression analysis of the transcriptomic data was performed using DeSeq2 (*Love et al., 2014*). Genes with a median of expression lower than 1 CPM, and genes containing outlier observations were filtered before statistical modeling, and the remaining N=4004 genes were used for differential expression analysis. p-Values were corrected for multiple testing using the R package, implementing the Storey-Tibshirani FDR correction method (*Storey et al., 2023*). Genes with an FDR <0.05 were deemed as differentially expressed. Data were deposited at Gene expression Omnibus (GEO); GSE143996.

## Quantitative real-time PCR

For reverse transcription cDNA was obtained from 10 µl of isolated RNA. Each reaction included an RT control, where RNA was replaced by 10 µl of DEPC-treated water. Reaction mixture 1 (volumes refer to a single sample) was prepared, containing 2 µl of random primers 500 ng/µl; 2 µl of dNTPs mix 10 mM; and 12.5 µl of DEPC-treated water. A total of 16.5 µl of this mixture was added to 10 µl of each sample, and the samples were incubated for 5 min at 65 °C to remove secondary structures. After 2 min on ice, samples were centrifuged at 12,000 rpm for 5 min. Reaction mixture 2 (volumes refer to a single sample) was prepared by mixing 8 µl of 5 X reverse transcription buffer; 2 µl of 0.1 M DTT; 2 µl of RNAseOUT inhibitor; and 1.5 µl of SuperScript IV reverse transcriptase 200 U/µl. This reverse transcriptase was chosen for its high specificity in synthesizing cDNA, particularly for nucleic acids with high G+C content, as is the case with *M. tuberculosis*. A total of 13.5 µl of mixture 2 was added to each sample, obtaining a final volume of 40 µl. The samples were incubated for 10 min at 25 °C and then at 55 °C for 30 min to synthesize cDNA. Finally, the reaction was stopped by inactivating the enzyme for 10 min at 80 °C and centrifuging briefly to recover the full volume. The samples were stored at –20 °C until use.

Amplification reactions were conducted by preparing a PCR reagent mixture containing, per sample: 1 µl of MgCl2; 1 µl of SYBR Green (previously mixed with L.C. Fast Start containing the polymerase); 0.25 µl of oligonucleotides specific to the sequences of interest at a concentration of 20 µM; and 4.5 µl of H2O. In this case, the sequences of interest are the Rv0432 (Forward primer: GCGTCTAC AGTTCCGGGTAC; Reverse primer: AGGGTACTGGTCAGGCTCTG) and Rv0433 (Forward primer: GCTGATCTGGGGTGTACACG; Reverse primer: GCCAACAGATGCGGGTAGTA) genes (Table supplement 1). The reaction was performed in a Light Cycler using capillaries containing 7 µl of the PCR mixture and 3 µl of each sample. For the negative control, the 3 µl were replaced with water, and for the positive control, they were replaced with 0.1 ng/µl DNA from the reference strain of *M. tuberculosis*. After a brief centrifuge pulse, the volume was introduced into the capillaries before analysis in the Light Cycler. Gene fold induction normalized to 16 S rRNA was calculated using the $2^{-\Delta\Delta Ct}$ method.

## Label-free mass spectrometry analysis

Total cell proteins and MEV-associated proteins were submitted to LC-MS label-free analysis using a Synapt G2Si ESI Q-Mobility-TOF spectrometer (Waters) coupled online to a NanoAcquity nano-HPLC (Waters), equipped with a Waters BEH C18 nano-column (200 mm x 75 um ID, 1.8 um). Samples were incubated and digested following the filter-aided sample preparation (FASP) protocol (*Wiśniewski et al., 2009*) Trypsin was added to a trypsin:protein ratio of 1:50, and the mixture was incubated overnight at 37 °C,dried out in a RVC2 25 speedvac concentrator (Christ), and resuspended in 0.1% FA. Peptides were desalted and resuspended in 0.1% FA using C18 stage tips (Millipore). Digested samples (500 ng) were loaded onto the LC system and resolved using a 60 min gradient (5 to 60% ACN). Data was acquired in HDDA mode that enhances signal intensities using the ion mobility separation step. Protein identification and quantification were carried out using Progenesis LC-MS (version 2.0.5556.29015, Nonlinear Dynamics). One of the runs was used as the reference to which the precursor masses in all other samples were aligned to. Only features comprising charges of 2+and 3+were selected. The raw abundances of each feature were automatically normalized and

logarithmized against the reference run. Samples were grouped in accordance with the comparison being performed, and an ANOVA analysis was performed. A peak list containing the information of all the features was generated and exported to the Mascot search engine (Matrix Science Ltd.). This file was searched against a Uniprot/Swissprot database, and the list of identified peptides was imported back to Progenesis LC-MS. Protein quantitation was performed based on the three most intense non-conflicting peptides (peptides occurring in only one protein), except for proteins with only two non-conflicting peptides.

## Proteomics data analysis

Normalization and imputation of the proteomics data was done using the DEP package for R (*Zhang et al., 2018*). Normalization was done using the vsn method (*Huber et al., 2002*), which transforms data tackling heteroskedasticity. Imputation of missing values was performed using the bpca method (*Oba et al., 2003*), which leans on principal component analysis to impute missing values.

Data was analyzed according to a design where differences between strains ($virR^{mut}$ vs -WT, as well as $virR^{mut}$ vs $virR^{mut}$-Comp) were estimated independently within each cell compartment, (whole cell lysates or EVs). according to a nested design: Expression ~Compartment + Strain:Compartment.

Differential expression analysis of the proteomics data was performed using limma (*Ritchie et al., 2015*) and p-values were corrected for multiple testing according to the Storey-Tibshirani method using the R package qvalue (*Storey et al., 2023*; *Storey and Tibshirani, 2003*). Proteins with FDR < 0.05 were selected as differentially expressed.

## Gene ontology (GO) enrichments

Enrichment analyses of the sets of differentially expressed genes and proteins were performed in R using the terms of the Gene Ontology database for testing, version 11.02.2020. Gene sets corresponding to the biological process and cellular component ontologies between levels 4–6 (N=2245) were selected and filtered according to their size (minimum size = 10 in the RNAseq analyses; and = 3 in the proteomics analyses). The selected sets were tested for enrichment using Fisher's exact test among genes, or proteins consistently up, or down regulated in $virR^{mut}$ with respect to both $virR$-expressing strains in each case. Multiple testing correction was performed using the qvalue R package (*Storey et al., 2023*). Terms with an FDR <0.1 and an odds ratio >2 were selected for visualization in *Figures 3E and 6F*, where node sizes were proportional to the significance of the enrichments. In these visualizations, two terms were connected depending on the amount of sharing between the groups of genes contributing to the enrichments in each of them, drawing a link whenever the intersection between the groups of genes was larger than 50% of the smallest of the two gene sets involved. Ontology term clusters, represented by colors, were then assigned using the Louvain's algorithm for modularity optimization (*Blondel et al., 2008*), as implemented in the R package igraph (*Csardi and Nepusz, 2006*). Visualization of the resulting networks was done using the open-source software Gephi (*Bastian et al., 2009*).

## Targeted metabolomics

LC-MS analysis was carried out on a Schimazdu LC/MS-8030 coupled with a triple quadrupole mass analyser (QqQ) provided with an electrospray source in a negative ionization mode. Extracted metabolites were quantified in Multiple Reaction Monitoring (MRM) mode. A stock standard solution prepared in 17% (v/v) methanol in water and containing all metabolites: acetyl-CoA, methyl citrate, methylmalonyl-CoA, pyruvate, was used as external standard. Metabolites were identified by ion pairing liquid chromatography. Acetyl-CoA, methyl citrate and methylmalonyl-CoA were separated by reverse phase with a poroshell 120 Phenyl-Hexyl analytical column (2.1x50 mm, 2.7 μm, Agilent) and a binary gradient (*Supplementary file 7*) consisting of a mobile phase A with water-methanol 97:3 (v/v), 10 mM tributylamine and 3 mM acetic acid, and mobile phase B composed of pure methanol. Pyruvate was separated by normal phase using a Kinetex HILIC analytical column (2.1x150 mm, 2.6 μm, Phenomenex) and a binary gradient (*Supplementary file 8*) consisting of pure water as mobile phase A and pure methanol as mobile phase B.

The flow rate was maintained at 0.4 ml/min and the injection volume was 20 μl. Nitrogen was used as nebulizing and drying gas at a flow rate of 1.5 and 15 ml/min, respectively. The desolvation line (DL) temperature was set at 250°C and the ionization voltage was fixed at 4.5 kV. Argon was used as

collision gas to perform collision-induced dissociation at a pressure of 230 kPa. The dwell time was set at 100ms and the detection was performed in MRM mode (detector voltage of 1.8 Kv), monitoring three transitions for each compound. The transition with higher intensity was selected as quantitative purposes (*Supplementary file 9*), while the other two were used for confirmation of the identity.

## Lipidomic analysis by UPLC-MS

The effect of virR on relative abundance of different cell wall lipids was characterized as had been done previously for MEVs (*Prados-Rosales et al., 2014b*). Briefly, cell wall lipids were extracted from triplicate 10 ml cultures of wild type, *virR*$^{mut}$, and *virR*$^{mut}$ complemented (*virR*$^{mut}$-C) strains overnight into 3 ml 2:1 chloroform:methanol at room temperature, extracts were dried under nitrogen before being dissolved in 0.75 ml LC-MS grade 2:1:1 isopropanol:acetonitrile:water. Extracts were separated on a Waters ACQUITY BEH C18 column UPLC heated to 55 °C and eluted at a flow rate of 0.4 ml/min with 60:40 acetonitrile:water as mobile phase A and 90:10 isopropanol:acetonitrile as mobile phase B, both containing 10 mM ammonium formate and 0.1% formic acid. Initial condition was 40%, B which increased to 43% at 2 min, 50% at 2.1 min, 54% at 12 min, 70% at 12.1 min, 90% at 18 min before returning to initial condition at 18.1 min to complete the run at 20 min. The coupled Waters Synapt G2 q-TOF MS was operated in positive ion resolution mode with the following conditions: capillary voltage 2kV, cone voltage 30 V, desolvation gas temperature 550 °C, gas flow 900 L/hr, source temperature 120 °C. Mass spectra were acquired in centroid mode from 100 to 3000 m/z with scan times of 0.5 s. Leucine enkephaline was used as reference. Relative abundances of Mass-retention time pairs were normalized to total ion current and lipid species were identified using Mycomass (*Layre et al., 2011*) and Mtb LipidDB databases (*Sartain et al., 2011*).

## Lipidomic analysis by thin layer chromatography (TLC)

Lipidomic analysis of both whole cells and EVs by TLC was performed as previously described (*Boldrin et al., 2021*). *Mtb* strains were grown at 37 °C in standing Middlebrook 7H9 and when they reached an OD595 of 0.2, cells were washed and transferred to MM. Cells were grown until they reached an OD595 of 0.5. Further, cells were harvested by centrifugation at 3500 rpm for 5 min, and pellets were heat-inactivated at 100 °C for 1 hr and used for lipid extraction. Five milliliters 2:1 (vol/vol) chloroform-methanol was added to the cell pellet and incubated at room temperature (RT) for 12 hr with constant stirring. The organic extract was separated by centrifugation (1000 × *g*, 5 min, RT) and decantation. The obtained pellet was extracted with 5 ml 2:1 (vol/vol) chloroform-methanol for 1 hr at RT with constant stirring and separated under the same conditions. The combined two organic extracts (total lipids) were dried under a stream of nitrogen gas at RT and saved for lipid analysis.

Polar lipids were separated by 2D-TLC by spotting 200 µg of total lipids on 60 F254 silica gel plates (Merck) using chloroform, methanol, water (50:30:6) as a mobile phase 1 (first dimension), and chloroform, acetic acid, methanol, and water (40:25:3:6) as mobile phase 2 (second dimension). Analysis of PDIMs and TAGs was performed in a 1D-TLC format. Two hundred µg of total lipids were spotted on the silica plates and separated using 9:1 petroleum ether-diethyl ether as mobile phase. Free mycolates were separated by 1D-TLC using chloroform/methanol/ammonium hydroxide (80:20:2) as mobile phase. Mycolic acids were extracted and separated from cells as previously described (*Vilchèze and Jacobs, 2007*). Sulfolipids (SLs) and diacyltrehaloses (DATs) were separated chloroform/methanol/water (100:14:0.8) (first dimension) and chloro- form/acetone/methanol/water (50:60:2.5:3) (second dimension). EV-associated lipids were separated and visualized following the same procedures as for WCLs. Development of TLC plates was performed by repeating the process of pulverizing the staining solution and heating the plate at 100 °C 4 times. The staining solution contained 10.5 ml 15% ethanolic solution of 1-naphthol, 6.5 ml 97% sulfuric acid, 40.5 ml ethanol, and 4 ml water.

## PG isolation

To obtain mycobacterial cell walls, whole cell pellets were resuspended in PBS and boiled for 15 min. Boiled cells were transferred to 2 ml Lysing matrix tubes with 0.1 mm silica beads (M.P. Biomedical, Santa Ana, CA, United States). And mechanically disrupted in a FastPrep-24TM (M.P. Biomedical) giving 12 cycles at predetermined speed of 6 during 30 s. Samples were allowed to chill on ice for 3 min between cycles. Cell breakage was monitored by optical microscopy (using Methylene blue as stain), if 95% of the cells had not broken perform additional FastPrep-24TM cycles were performed.

Lysed cells were centrifuged at 170×g for 30 s to separate them from beads and the supernatant were transferred to a clean 1.5 ml tube. The suspensions were further digested with 10 µg of DNase and RNase/ml for 1 h at 4 °C to obtain a cell wall-enriched fraction after centrifugation at 27,000×g for 10 min. Cell walls were resuspended in PBS containing 2% sodium dodecyl sulfate (SDS) and the suspension was incubated for 1 hr at 90°C with constant stirring and recentrifuged at 27,000×g for 30 min, and the supernatant was discarded. This process was repeated twice. The resulting pellet was washed trice in distilled water, 80% acetone and acetone and lyophilized. Pellets were resuspended in 0.9 ml Tris-HCL (50 mM, pH 7) buffer, plus 0.1 ml of protease K stock solution. The mixture was incubated at 60 °C for 90 min, to digest any remaining surface proteins attached to the CW and centrifuged again for 3 min at 20,000×g and resuspend pellet in distilled water. The mixture was then heated at 90 °C for 1 h before centrifugation at 27,000×g for 30 min. The supernatant was discarded and washed twice with PBS and four times with deionized water to remove SDS. Pure PG was obtained from isolated cell walls. Briefly, pure cell walls were resuspended in 0.5% KOH in methanol and stirred at 37 °C for 4 days to hydrolyze mycolic acids (MA). The mixture was centrifuged (27,000×g 20 min) and the pellet was washed twice with methanol. MA were separated from cell walls using diethyl ether and recentrifuged at 27,000×g for 20 min. The supernatant contained MAMEs. The extraction process with diethyl ether was repeated twice and let dry. The resulting arabinogalactan (AG)-PG was digested with 0.2 N $H_2SO_4$ at 85 °C for 30 min and neutralized with $BaCO_3$ to remove the arabinogalactan. Treated AG-PG was centrifuged at 27,000×g for 20 min. Supernatant contained soluble AG. The resulting insoluble PG was washed four times by centrifugation in deionized water and stored at 4 °C or room temperature.

## Analysis of diaminopimelic acid

For DAP quantification (*Tsuruoka et al., 1985*), samples were hydrolyzed overnight in 6 N HCl at 100 °C in 1 ml capacity-crystal ampoules (Wheaton). Samples were dried and resuspended in water and mixed in a 1:1:1 ratio with pure acetic acid and ninhydrin reagent (250 mg ninhydrin, 4 ml phosphoric acid 0.6 M, and 6 ml pure acetic acid). Samples were measured at OD 434 nm, and the amounts of DAP were calculated based on a standard curve with pure DAP. The data displays the average ± standard deviations n=3.

## Glycosyl composition of isolated cell walls

500 µg of purified cell walls were mixed with 10 µg of inositol and hydrolyzed with 0.1 M trifluoroacetic acid (TFA) for 2 hr at 121 °C. The monosaccharides released were then converted into alditol acetates by reduction with $NaBH_4$ and acetylation with acetic anhydride in pyridine. The products were analyzed by gas chromatography-mass spectrometry in a 7890 A/5975 C system from Agilent, using a DB-5ht (30 m x 0.25 mm x 0.1 µm) fused silica capillary column. The compounds were identified by comparing the retention times of the peaks with those recorded for standards analyzed under identical conditions. Quantification was carried out considering the area of the peaks and the response factors of each compound with respect to inositol.

## Atomic force microscopy and data analysis

The dry pellets of purified peptidoglycan from the three strains were dissolved in pure water and further broken by tip sonication at 2 mA for 30 s, three cycles. Then 10 µl of the PG suspension was added to standard AFM substrate made of mica coated with 0.01% Poly-L-lysine by incubating the sample during 90 min to achieve good coverage. The sample was further rinsed with water and dried with Nitrogen flow. The AFM in ambient conditions (i.e. air) was performed using Tapping mode in a Bruker FastScan Bio machine (Santa Barbara, CA, United States) with Nunano SCOUT 350 - Silicon AFM probes (spring constant: 42 N/m, Resonance frequency: 350 kHz). The images were taken using a free amplitude of 10 nm with set point of 70–80% of free amplitude (e.g. 7 nm). For AFM high-resolution imaging in liquid the images were acquired in Peak force Tapping mode in imaging buffer made of 10 mM Tris and 150 mM KCl, pH: 8, with the FastScan Bio machine using Bruker Fastscan-D AFM probes (spring constant: 0.1 N/m, Resonance frequency: 70 kHz) at the force range of 1–3 nN peak force set point. The imaging parameters used are as follows; Scan rate: 1 Hz; Scan angle: 0 °; Peak force amplitude: 80–100 nm and with high pixel resolution (512-880).

The thickness measurements were performed with standard Gwyddion 1D height distribution tool (selecting an area containing background and a single leaflet of peptidoglycan). The number of cells analysed varied from n=23 (for Mtb WT), n=36 (for *virR^mut*), and n=25 (for *virR^mut*-C). The comparisons extracted from this data is statistically relevant with t test with Welch's correction performed. The analysis of the high-resolution images obtained in liquid was performed using the semi-automated custom-made routine from (Pasquina-Lemonche, et al. 'Pore AFM', 2024) where the pore area, later converted in diameter, and the pore number were measured from the 2D slice of each image containing the maximum number of pores (see *Figure 8—figure supplement 1*). The number of images similar to images shown in *Figure 8* were n=8 for all strains, analyzing both pore size and pore number, the statistical comparisons were also performed using a t test with Welch's correction.

## Blue native PAGE

Oligomerization of VirR and Lcp1 was analyzed by Blue native PAGE (Invitrogen, Waltham, MA, United States). Recombinant VirR and Lcp1 proteins were prepared in 1xLB Native PAGE including DDM at 0.5% final concentration. After incubation at 4 °C for 30 min, samples were centrifuged at 20,000×*g* for 30 min at 4 °C. The supernatant was recovered and mixed with G-250 additive following manufacturer instructions (Invitrogen) and loaded onto a Novex Bist-Tris 4–16% gradient gel and run following manufacturer instructions. Gel was first fixed and then stained with Coomassie.

## Liposome flotation assay

Individual or combined VirR and Lcp1 were prepared at 2 µM in buffer A in a final volume of 50 µl and incubated for 30 min at 37 °C. This volume was mixed with 100 µl of 60% optiprep (Sigma-Aldrich) to a final concentration of 40% and overlaid with 100 µl of subsequent optiprep solutions at 35%, 30%, 20%, and 10% (optiprep solutions were prepared in Buffer A). The samples were centrifugated at 70,000 rpm for 2 hr at 4 °C in an Optima TLX Ultracentrifuge (Beckman Coulter, Indianapolis, IN United States). Eleven fractions of 50 µl were recovered and examined by dot blot (performed as previously described). Polyclonal antibodies raised against VirR and Lcp1 were used as primary antibodies. Quantification of dotblots was performed in ImageJ as previously described (*Schirmer et al., 2022*). Briefly, integrated densities were calculated after background subtraction, and referred to wild type values to obtain the relative integrated intensities.

## Lysozyme and PG and AG susceptibility assays

*Mtb* WT and *virR^mut* strains were grown in 7H9 media supplemented with ADC and Tween 80 to an OD 540 of 0.2 and then diluted to an OD540 of 0.1 with 7H9 containing 0 or 50 µgml⁻¹ lysozyme (Sigma). The cultures were incubated with agitation at 37 °C and OD540 was monitored daily.

## Fluorescence microscopy

The *Mtb* strain bearing the plasmid VirR-pMV261-Venus was grown in MM to an OD595nm of 0.4. At this stage, cells were harvested, washed three times in 1×PBS (pH 7.4), resuspended in 1/3 – 1/6 volume of 4% paraformaldehyde (PFA) and incubated for 2 hr at 37 °C. Finally, the cells were washed twice in 1×PBS and resuspended in 100–200 µl of PBS. Microscope slides were covered with 5 µl poly-L-lysine (Sigma) and excess poly-L-lysine was removed away with distilled water. A 20–40 µl aliquot was applied to pre-treated slides, allowed to air dry and covered with antifade solution (Prolong Gold, Invitrogen) and visualized under the fluorescence microscope using a 100 x oil immersion objective on an Orca 12 ERG digital CCD camera (Hamamatsu Photonics) mounted on a Nikon E600 microscope.

## FACS analysis for split-GFP experiments

*M. smegmatis* mc²155 transformed with the pGMCS derivatives were inoculated in complete 7H9 medium. After overnight incubation at 37 °C, bacteria were collected and analyzed by fluorescence activated cell sorting using a BD FACS LSRFortessa X20 flow cytometer. Flow cytometry data analysis was performed using FlowJo software (Version 10; Becton, Dickinson and Company, Ashland, OR, United States). The gating strategy is displayed in *Figure 9—figure supplement 2*.

## Statistical analysis

The statistical significance of the difference between experimental groups was determined by the two-tailed Student's test using PRISM 5.0. *P*-values ≤than 0.05 were considered significant. Statistical analysis of RNAseq is detailed in the corresponding section of Materials and Methods.

## Acknowledgements

We are very grateful to Carl Nathan for sharing mutant Mtb strains. We thank G Marcela Rodriguez for sharing the a-IdeR antibody. We thank Heran Darwin for sharing a-cei polyclonal antibody. We thank Emmanuelle Näser (Génotoul TRI-IPBS platform, Toulouse) for helping with flow cytometry. We thank Dr. Luke Alderwick (University of Birmingham) for the donation of the plasmid to recombinantly express Lcp1 protein. We thank M Gutiérrez for assistance with figures. The authors have no conflict of interest to declare. RP-R acknowledges support by MINECO/FEDER EU contracts SAF2016–77433-R, PID2019-110240RB-I00, PID2022-136611OB-I00; and NIH-R01AI162821. CICbioGUNE thanks MINECO for the Severo Ochoa Excellence Accreditation (SEV-2016–0644). LP-L and JKH acknowledges support by the Wellcome Trust (212197/Z/19/Z) which funded part of this research. For the purpose of open access, the authors have applied and will apply a CC BY public copyright license to any Author Accepted Manuscript version arising from this submission. JS acknowledges support from the Spanish Ministry of Science and Innovation (MICINN) through grant PID2019-106859GA-I00 and Ramón y Cajal research grant RYC-2017-23560, as well as to the Government of Aragón, Spain, through grant B49-23R (NeuroBioSys). FC lab research is supported by the Swedish Research Council (VR2018-02823 and VR2018-05882), the Laboratory for Molecular Infection Medicine Sweden (MIMS), Umeå University, the Knut and Alice Wallenberg Foundation (KAW) grant KAW2012.0184 and the Kempe Foundation.

## Additional information

### Funding

| Funder | Grant reference number | Author |
| --- | --- | --- |
| Ministerio de Economía y Competitividad | SAF2016-77433-R | Rafael Prados-Rosales |
| Ministerio de Economía y Competitividad | PID2019-110240RB-I00 | Rafael Prados-Rosales |
| Ministerio de Economía y Competitividad | PID2022-136611OB-I00 | Rafael Prados-Rosales |
| National Institutes of Health | R01AI162821 | Rafael Prados-Rosales |
| Ministerio de Ciencia e Innovación | PID2019-106859GA-I00 | Joaquín Sanz |
| Wellcome Trust | 212197/Z/19/Z | Jamie K Hobbs<br>Laia Pasquina-Lemonche |
| Swedish Research Council | VR2018-02823 | Felipe Cava |
| Swedish Research Council | VR2018-05882 | Felipe Cava |
| Knut and Alice Wallenberg Foundation | KAW2012.0184 | Felipe Cava |
| Kempe Foundation | | Felipe Cava |
| Government of Aragón, Spain | B49-23R (NeuroBioSys) | Joaquín Sanz |
| Ministerio de Economía y Competitividad | Severo Ochoa Excellence Accreditation (CICbioGUNE) SEV-2016–0644 | Ainhoa Palacios<br>Felix Elortza<br>Mikel Azkargorta |

| Funder | Grant reference number | Author |
| --- | --- | --- |
| Ministerio de Ciencia e Innovación | RYC-2017-23560 | Joaquín Sanz |

The funders had no role in study design, data collection and interpretation, or the decision to submit the work for publication. For the purpose of Open Access, the authors have applied a CC BY public copyright license to any Author Accepted Manuscript version arising from this submission.

## Author contributions

Vivian C Salgueiro-Toledo, Claude Gutierrez, Data curation, Formal analysis, Validation, Investigation, Methodology, Writing – original draft; Jorge Bertol, Data curation, Formal analysis, Investigation, Methodology, Writing – original draft, Writing – review and editing; Jose L Serrano-Mestre, Lucia Vázquez-Iniesta, Laia Pasquina-Lemonche, Akbar Espaillat, Brian Weinrick, Data curation, Formal analysis, Investigation, Methodology; Noelia Ferrer-Luzon, Ainhoa Palacios, Laura Lerma, Formal analysis, Investigation, Methodology; Jose L Lavin, Data curation, Software, Formal analysis; Felix Elortza, Supervision, Validation, Methodology; Mikel Azkargorta, Formal analysis, Investigation, Methodology, Writing – original draft; Alicia Prieto, Investigation, Methodology, Writing – original draft; Pilar Buendía-Nacarino, Investigation, Methodology; Jose L Luque-García, Supervision, Validation, Investigation, Writing – original draft; Olivier Neyrolles, Supervision, Funding acquisition, Validation, Investigation, Writing – original draft; Felipe Cava, Supervision, Funding acquisition, Validation, Investigation; Jamie K Hobbs, Resources, Supervision, Validation, Writing – original draft; Joaquín Sanz, Data curation, Software, Formal analysis, Supervision, Funding acquisition, Investigation, Methodology, Writing – original draft, Project administration, Writing – review and editing; Rafael Prados-Rosales, Conceptualization, Formal analysis, Supervision, Funding acquisition, Validation, Investigation, Methodology, Writing – original draft, Project administration, Writing – review and editing

## Author ORCIDs

Claude Gutierrez ⓘ https://orcid.org/0000-0003-1777-0223
Brian Weinrick ⓘ https://orcid.org/0000-0003-0880-4487
Mikel Azkargorta ⓘ https://orcid.org/0000-0001-9115-3202
Jose L Luque-García ⓘ https://orcid.org/0000-0001-6273-0349
Felipe Cava ⓘ https://orcid.org/0000-0001-5995-718X
Joaquín Sanz ⓘ https://orcid.org/0000-0002-2980-9685
Rafael Prados-Rosales ⓘ https://orcid.org/0000-0001-5964-0166

Reviewer #1 (Public review): https://doi.org/10.7554/eLife.94982.4.sa1
Reviewer #2 (Public review): https://doi.org/10.7554/eLife.94982.4.sa2
Author response https://doi.org/10.7554/eLife.94982.4.sa3

# Additional files

## Supplementary files

Supplementary file 1. FileS1_RNA_differential_expression.xlsx: Results of the differential expression analyses for the RNA-seq dataset.

Supplementary file 2. FileS2_GOenrichments_RNA.xlsx: Gene ontology enrichment analyses conducted on sets of genes up (N=399), and down-regulated (N=502) in the *virR^mut^* strain. Rows highlighted correspond to terms selected for graphical representation (*Figure 3E*, OR >2 & FDR <0.1).

Supplementary file 3. FileS3_TRN_regulon_enrichments.xlsx: Enrichment analyses for the Transcription factor regulons reported in *Minch et al., 2015*.

Supplementary file 4. FileS4_Proteins_differential_expression.xlsx: Results of the differential expression analyses for the LF-MS dataset.

Supplementary file 5. FileS5_GOenrichments_proteins.xlsx: Gene ontology enrichment analyses conducted on sets of protein up (N=117), and down-regulated (N=108) in the extracellular vesicles of the *virR^mut^* strain. Rows highlighted correspond to terms selected for graphical representation

(*Figure 5F*, OR >2 & FDR <0.1).

Supplementary file 6. Plasmids and primers used in this study.

Supplementary file 7. Gradient mobile phase composition for ion-pairing LC-MS/MS for the quantification of acetyl-CoA, methyl citrate and methylmalonyl-CoA.

Supplementary file 8. Gradient mobile phase composition for ion-pairing LC-MS/MS for the quantification of pyruvate.

Supplementary file 9. Quantifier MRM transitions and collision energies used for acety-CoA, methyl citrate, methylmalonyl-CoA and pyruvate.

MDAR checklist

## Data availability

Sequencing data have been deposited in GEO under accession code GSE143996.

The following dataset was generated:

| Author(s) | Year | Dataset title | Dataset URL | Database and Identifier |
|---|---|---|---|---|
| Gupta S, Bhagavathula M, Sharma V, Sharma N | 2023 | Dynamin-like proteins are essential for vesicle biogenesis in *Mycobacterium tuberculosis* | https://www.ncbi. nlm.nih.gov/geo/ query/acc.cgi?acc= GSE143996 | NCBI Gene Expression Omnibus, GSE143996 |

The following previously published dataset was used:

| Author(s) | Year | Dataset title | Dataset URL | Database and Identifier |
|---|---|---|---|---|
| Pawełczyk J, Brzostek A, Minias A, Płociński P, Rumijowska-Galewicz A, Strapagiel D, Zakrzewska-Czerwińska J, Dziadek J | 2021 | Transcriptomic changes associated with growth of *Mycobacterium tuberculosis* H37Rv wild-type strain and ΔprpR mutant in mineral medium supplemented with cholesterol as the sole carbon source or standard medium supplemented with vitamin B12 | https://www.ncbi. nlm.nih.gov/geo/ query/acc.cgi?acc= GSE175812 | NCBI Gene Expression Omnibus, GSE175812 |

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
