## [Editor Report · eLife Assessment]

In this **important** study, the authors investigate the biogenesis of extracellular vesicles in mycobacteria and provide several observations to link VirR with vesiculogenesis, peptidoglycan metabolism, lipid metabolism, and cell wall permeability. The authors have done a commendable job of comprehensively examining the phenotypes associated with the VirR mutant using various techniques. The evidence presented in the revised manuscript is **convincing** and creates several avenues for further research.

---

## [Referee Report · Reviewer #1 (Public review)]

Summary:

The present study's main aim is to investigate the mechanism of how VirR controls the magnitude of MEV release in Mtb. The authors used various techniques, including genetics, transcriptomics, proteomics, and ultrastructural and biochemical methods. Several observations were made to link VirR-mediated vesiculogenesis with PG metabolism, lipid metabolism, and cell wall permeability. Finally, the authors presented evidence of a direct physical interaction of VirR with the LCP proteins involved in linking PG with AG, providing clues that VirR might act as a scaffold for LCP proteins and remodel the cell wall of Mtb. Since the Mtb cell wall provides a formidable anatomical barrier for the entry of antibiotics, targeting VirR might weaken the permeability of the pathogen along with the stimulation of the immune system due to enhanced vesiculogenesis. Therefore, VirR could be an excellent drug target. Overall, the study is an essential area of TB biology.

Strengths:

The authors have done a commendable job of comprehensively examining the phenotypes associated with the VirR mutant using various techniques. Application of Cryo-EM technology confirmed increased thickness and altered arrangement of CM-L1 layer. The authors also confirmed that increased vesicle release in the mutant was not due to cell lysis, which contrasts with studies in other bacterial species.

Another strength of the manuscript is that biochemical experiments show altered permeability and PG turnover in the mutant, which fits with later experiments where authors provide evidence of a direct physical interaction of VirR with LCP proteins.

Transcriptomics and proteomics data were helpful in making connections with lipid metabolism, which the authors confirmed by analyzing the lipids and metabolites of the mutant.

Lastly, using three approaches, the authors confirm that VirR interacts with LCP proteins in Mtb via the LytR_C terminal domain.

Altogether, the work is comprehensive, experiments are designed well, and conclusions were made based on the data generated after verification using multiple complementary approaches.

Weaknesses:

The major weakness is that the mechanism of VirR-mediated EV release remains enigmatic. Most of the findings are observational and only associate enhanced vesiculogenesis observed in the VirR mutant with cell wall permeability and PG metabolism. Authors suggest that EV release occurs during cell division when PG is most fragile. However, this has yet to be tested in the manuscript-the AFM of the VirR mutant, which produces thicker PG with more pore density, displays enhanced vesiculogenesis. No evidence was presented to show that the PG of the mutant is fragile, and there are differences in cell division to explain increased vesiculogenesis. These observations, counterintuitive to the authors' hypothesis, need detailed experimental verification.

Transcriptomic data only adds a little substantial. Transcriptomic data do not correlate with the proteomics data. It remains unclear how VirR deregulates transcription. TLCs of lipids are not quantitative. For example, the TLC image of PDIM is poor; quantitative estimation needs metabolic labeling of lipids with radioactive precursors. Further, change in PDIMs is likely to affect other lipids (SL-1, PAT/DAT) that share a common precursor (propionyl- CoA).

The connection of cholesterol with cell wall permeability is tenuous. Cholesterol will serve as a carbon source and contribute to the biosynthesis of methyl-branched lipids such as PDIM, SL-1, and PAD/DAT. Carbon sources also affect other aspects of physiology (redox, respiration, ATP), which can directly affect permeability and import/export of drugs. Authors should investigate whether restoration of the normal level of permeability and EV release is not due to the maintenance of cell wall lipid balance upon cholesterol exposure of the VirR mutant.

Finally, protein interaction data is based on experiments done once without statistical analysis. If the interaction between VirR and LCP protein is expected on the mycobacterial membrane, how SPLIT_GFP system expressed in the cytoplasm is physiologically relevant. No explanation was provided as to why VirR interacts with the truncated version of LCP proteins and not with the full-length proteins.

Comments on revisions:

The authors have addressed my comments. I have no further issues.

---

## [Referee Report · Reviewer #2 (Public review)]

Summary:

In this work, Vivian Salgueiro et al. have comprehensively investigated the role of VirR in the vesicle production process in Mtb using state-of-the-art omics, imaging, and several biochemical assays. From the present study, authors have drawn a positive correlation between cell membrane permeability and vasculogenesis and implicated VirR in affecting membrane permeability, thereby impacting vasculogenesis.

Strengths:

The authors have discovered a critical factor (i.e. membrane permeability) that affects vesicle production and release in Mycobacteria, which can broadly be applied to other bacteria and may be of significant interest to other scientists in the field. Through omics and multiple targeted assays such as targeted metabolomics, PG isolation, analysis of Diaminopimelic acid and glycosyl composition of the cell wall, and, importantly, molecular interactions with PG-AG ligating canonical LCP proteins, the authors have established that VirR is a central scaffold at the cell envelope remodelling process which is critical for MEV production.

Comments on the revision.

Authors have addressed the concerns, specifically regarding the expression of downstream genes. It appears that they are not altered significantly.

Data in Fig 6C shows significantly higher expresssion of VirR compared to control or knock down. In the absence of using a regulatable expression such as nitrile, this is expected from a constitutive promoter.

I have no further questions for the author.

---

## [Author Response]

The following is the authors’ response to the previous reviews.

**Comments on the revised version:**
Concerns flagged about using CRISPR -guide RNA mediated knockdown of viral has yet to be addressed entirely. I understand that the authors could not get knock out despite attempts and hence they have guide RNA mediated knockdown strategy. However, I wondered if the authors looked at the levels of the downstream genes in this knockdown.

We thank the reviewer for bringing this up since it is known that certain artifacts derived from this approach may be related with changes in expression of downstream genes. We run a qPCR of Rv0432 and Rv0433 and confirmed that no significant differences in expression of *virR* downstream genes were detected in the *virR* mutant or the complemented strains relative to WT. This is now indicated in the method section on Generation of the CRISPR mutants. The data is now presented as Supplementary Figure 13.

Authors have used the virmut-Comp strain for some of the experiments. However, the materials and methods must describe how this strain was generated. Given the mutant is a CRISPR-guide RNA mediated knockdown. The CRISPR construct may have taken up the L5 loci. Did authors use episomal construct for complementation? If so, what is the expression level of virR in the complementation construct? What are the expression levels of downstream genes in mutant and complementation strains? This is important because the transcriptome analysis was redone by considering complementation strain. The complemented strain is written as virmut-C or virmut-Comp. This has to be consistent.

We apologize for not having included the information about the generation of the complemented strain in our last version of the manuscript. We took the complementing vector from a previous paper on VirR (Rath et al., (2013) PNAS 110(49):E4790). This vector was constructed as follows: Complementation plasmids were cloned using Gateway Cloning Technology (Invitrogen). *E. coli* strains expressing the following Gateway vectors were kindly provided by Dirk Schnappinger and Sabine Ehrt: pDO221A, pDO23A, pEN23A-linker1, pEN41A-TO2, pEN21A-Hsp60, pDE43-MEH. PCR was used to amplify the following target sequences from H37Rvgenomic DNA: coding sequence of Rv0431, coding sequence of Rv0431 with a FLAG tag either in its C-terminus or its N-terminus, and the predicted cytosolic sequence of Rv0431 with a FLAG tag in its new C-terminus. The primers used for PCR were designed such that the amplicons would be flanked with Gateway cloning- specific attachment (att) sites. These PCR products were recombined into Gateway donor vectors using bacteriophage-derived integrase and integration host factor, resulting in entry vectors. The recombination events are specific to the attB sites on the PCR products and to the attP sequences on the donor vectors, such that the orientation of the target sequence is maintained during the recombination reaction, also known as the BP reaction, for attB-attP recombination. Using the MultiSite Gateway system, three DNA fragments, derived from each of three distinct entry vectors, can be simultaneously inserted into a final complementation vector called the destination vector in a specific order and orientation. Multisite recombination events are mediated by Integrase and Integrase Host Factor, in a process called the LR reaction (for the attL and attR sites in the entry and destination vectors). The Gateway entry vectors thus generated were recombined with another entry vector containing either the Hsp60 promoter, an empty entry vector, and a complementation vector (episomal) to give rise to the final destination vector. The destination vector (episomal) was engineered to contain a hygromycin resistance cassette. These vectors were used to transform competent Rv0431-deficient *Mtb*. The transformation mixture was plated on 7H11 plates containing OADC and hygromycin (50 μg/ml). Colonies, typically observed 3-5 weeks later, were isolated and grown in 7H9 media and characterized.

For simplicity, we have just referenced our previous paper to indicate that the complementing plasmid is the same used in that study.

Regarding the *virR* expression levels in the WT, *virR^mut^* and complemented *virR* strains please see previous Figure 6 C.

**Recommendations for the authors:**

**Reviewer #1 (Recommendations for the authors):**
The authors have revised the manuscript in light of previous reviews. The authors have addressed some of my concerns appropriately. However, the specific dataset remains unchanged and unclear.Fig 8G and H: In response to a comment on the mechanism of how VirR mediates EV release, the authors have added new data showing an increase in the abundance of deacetylated muropeptides in the mutant. This observation is linked to altered lysozyme activity or PG fragility. In my opinion, this is another indirect observation. More concerning is the complemented strain, which also showed a comparable increase in deacetylated muropeptides, indicating that the altered muropeptides could be unrelated to VirR.

We must disagree here with the reviewer assessment about the fact that the abundance of deacetylated muropeptides is an indirect indication of PG fragility. We consider that this observation and quantitative fact is another additional evidence that indicate a more fragile PG. We believe that considering each of the supporting facts individually may be seen as indirect, but we would like that the reviewer take all the evidence together: (i) sensitivity to lysozyme; (ii) enlargement and altered physicochemical morphological characteristics including porosity or thickness; (iii) altered penetrance of FDAAs; and (iv) increased released of muropeptides. In this later fact, the complemented strain may not display the WT features, but this may be due to some artifacts derived from the complementation.

Taking all together, we believe that the PG of *virR^mut^* is more fragile than that of the WT and the complemented strains based on a series of evidence. We hope the reviewer may consider this perspective when analyzing such a complex feature like PG fragility. So far, there is not a direct method to assess this condition.

Lipid analyses are not comprehensive. The issue related to the need for more clarity of DIMA and DIMB still needs to be addressed. I understand that the authors do not have facilities to perform radioactive assays. However, they could have repeated the experiment to generate a better-quality image. Similarly, the newly generated SL-1, PAT, and DAT TLC could be of better quality. Bands still need to be resolved. The solvent front is irregular. The same is true for PIMs and DPG TLCs. With the evidence provided, the deregulation of cell wall lipids is incomplete.

We agree with the reviewer that the quality of the TLC is not appropriate. We have no repeated the PDIM TLC (new Fig 7D). In addition, we have repeated the TLCs resolving sulfolipids in a 2D mode. For simplicity we just run the glycerol condition including the three strains. This is now part of a new Supplementary figure 8 B. For PIMs, we have a 1D and a 2D analysis that, after checking previous papers using similar approaches with no radioactivity, we consider that it has the desired quality to identify the indicated lipids.

We hope this new data and repeated experiments satisfy the reviewer concerns.

Thank you very much for your assessment and time to review this paper.